# PatchMixer: A Patch-Mixing Architecture for Long-Term Time Series Forecasting

## Abstract

Although the Transformer has been the dominant architecture for time series forecasting tasks in recent years, a fundamental challenge remains: the permutation-invariant self-attention mechanism within Transformers leads to a loss of temporal information. To tackle these challenges, we propose PatchMixer, a novel CNN-based model. It introduces a permutation-variant convolutional structure to preserve temporal information. Diverging from conventional CNNs in this field, which often employ multiple scales or numerous branches, our method relies exclusively on depthwise separable convolutions. This allows us to extract both local features and global correlations using a single-scale architecture. Furthermore, we employ dual forecasting heads that encompass both linear and nonlinear components to better model future curve trends and details. Our experimental results on seven time-series forecasting benchmarks indicate that compared with the state-of-the-art method and the best-performing CNN, PatchMixer yields 3.9% and 21.2% relative improvements, respectively, while being 2-3x faster than the most advanced method. We will release our code and model.

## 1    Introduction

Long-term time series forecasting (LTSF) is a crucial task aimed at predicting future trends over an extended period by leveraging substantial historical time-series data. LTSF applications span a wide range of domains, including traffic flow estimation, energy management, and financial investment.

Transformer (Vaswani et al., 2017) has been the dominant architecture in time series forecasting tasks in the last few years. It was first applied in the field of Natural Language Processing (NLP) and later extended as a universal architecture to the field of Computer Vision (CV) and so on. To address the limitations of the vanilla Transformer models, such as quadratic time or memory complexity, Informer (Zhou et al., 2021) introduced an innovative Transformer architecture with reduced complexity.Subsequently, numerous Transformer variants (Wu et al., 2021; Zhou et al., 2022; Liu et al., 2022b) emerged in the field of time series analysis, to enhance performance or improve computational efficiency.

However, the effectiveness of Transformers in LTSF tasks has been called into question by an experiment involving simple Multi-Layer Perception (MLP) networks (Zeng et al., 2023), which surprisingly surpassed the forecasting performance of all previous Transformer models. Therefore, they posed an intriguing question: Are Transformers effective for long-term time series forecasting? In response to this, a Transformer-based model, PatchTST (Nie et al., 2023), used a patch-based technique motivated by CV and reached state-of-the-art (SOTA) prediction results. Recent transformers (Zhang & Yan, 2023; Lin et al., 2023) also adopted patch-based representations and achieved noteworthy performance. This naturally gives rise to another important question: Does the impressive performance of PatchTST primarily stem from the inherent power of the Transformer architecture, or is it, at least in part, attributed to the use of patches as the input representation?

In this paper, we address this issue by introducing a novel backbone architecture called **PatchMixer**, which is based on Convolutional Neural Networks (CNNs). PatchMixer is primarily composed of two convolutional layers and two forecasting heads. Its distinguishing feature is the "patch-mixing" design, which means the model initially segments the input time series into smaller temporal patches and subsequently integrates information from both within and between these patches. Motivated by the multi-head attention mechanism in Transformer, we employ the dual forecasting heads design

in our model. These enhancements enable PatchMixer to outperform other CNN-based models, leading to state-of-the-art accuracy in time series forecasting.

The main contributions of this work are as follows:

- We propose PatchMixer, a novel model built on convolutional architecture. This approach efficiently replaces the computation-expensive self-attention module in Transformers while leveraging a novel patch-mixing design to uncover intricate temporal patterns in the time series.

- PatchMixer is efficient in long-term time series forecasting. By adopting a single-scale structure and optimizing patch representations, our model achieves a significant performance boost. It is 3x faster for inference and 2x faster during training compared to current SOTA model.

- On seven popular long-term forecasting benchmarks, our PatchMixer outperforms the SOTA method by $3.9\%$ on MSE and $3.0\%$ on MAE. Besides, our model achieves a substantial $21.2\%$ relative reduction in MSE and $12.5\%$ relative reduction in MAE on average compared with the previous best CNN model.

## 2 RELATED WORK

**CNNs for long-term context.** CNNs typically employ a local perspective, with layered convolutions extending their receptive fields across input spaces. For instance, TCN (Bai et al., 2018) first introduced CNN structure into TSF tasks, employing causal and dilated convolutions for temporal relationships. SCINet (Liu et al., 2022a) furthered this by extracting multi-resolution information through a binary tree structure. Recently, MICN (Wang et al., 2023) adopted multi-scale hybrid decomposition and isometric convolution for feature extraction from both local and global perspectives. TimesNet (Wu et al., 2023) segmented sequences into patches cyclically for temporal pattern analysis using visual models like Inception (Szegedy et al., 2015). On the single-scale front, S4 (Gu et al., 2021) processed long sequences through structured state spaces, offering valuable insights into long-term dependency modeling. CKConv (Romero et al., 2021) leveraged continuous convolutional kernels cater to varied data. Hyena (Poli et al., 2023) suggests adaptability to model long-term contexts, by using a combination of long convolutions and gating.

**Depthwise Separable Convolution.** This is a widely employed technique used in the field of computer vision. The work of depthwise separable convolutions was initially unveiled (Sifre & Mallat, 2014) in 2014. Later, this method was used as the first layer of Inception V1 (Szegedy et al., 2015) and Inception V2 (Ioffe & Szegedy, 2015). During the same period, Google introduced an efficient mobile model, called MobileNet (Howard et al., 2017). Its core layers were built on depthwise separable filters. Consequently, the Xception (Chollet, 2017) network demonstrated how to scale up depthwise separable filters. Recently, ConvMixer (Trockman & Kolter, 2022) via the method suggested the patch representation itself may be a critical component to the "superior" performance in CV tasks.

**Channel Independence.** A multivariate time series can be seen as a signal with multiple channels. When the input tokens take the vector of all time series features and project it to the embedding space to mix information, it is called "channel mixing". "channel independence" is exactly the opposite. Intuitively, the correlation among variables may help improve prediction accuracy. Zeng et al. (2023) used this strategy for the first time in the LTSF field, and its effectiveness was further verified in Nie et al. (2023). These two studies have shown that strategies emphasizing channel independence are more effective than channel mixing methods for forecasting tasks. Therefore, we adopt a channel-independent approach instead of a channel-mixing design. Furthermore, motivated by this concept, we explore correlations between and within patches of each univariate time series, which aligns with the idea of "patch mixing".

## 3 PROPOSED METHOD

### 3.1 PROBLEM FORMULATION

In this work, we address the following task: Given a set of multivariate time series instances with a historical look-back window $L : (\boldsymbol{x}_1, \ldots, \boldsymbol{x}_L)$, where each $\boldsymbol{x}_t$ at time step $t$ represents a vector of $M$ variables. Our objective is to make predictions for the subsequent $T$ time steps, resulting in the prediction sequence $(\boldsymbol{x}_{L+1}, \ldots, \boldsymbol{x}_{L+T})$.

From the perspective of channel independence, the multivariate time series $(\boldsymbol{x}_1, ..., \boldsymbol{x}_L)$ is split to $M$ univariate series $\boldsymbol{x}^{(i)} \in \mathbb{R}^{1 \times L}$. We consider the $i$-th univariate series of length $L$ as $\boldsymbol{x}_{1:L}^{(i)} = (\boldsymbol{x}_1^{(i)}, ..., \boldsymbol{x}_L^{(i)})$ where $i = 1, ..., M$. These univariate series are independently fed into the model. At last, the networks provide prediction results $\hat{\boldsymbol{x}}^{(i)} = (\hat{\boldsymbol{x}}_{L+1}^{(i)}, ..., \hat{\boldsymbol{x}}_{L+T}^{(i)}) \in \mathbb{R}^{1 \times T}$ accordingly.

### 3.2 MODEL STRUCTURE

The overall architecture of PatchMixer is illustrated in Figure 1. We employ a single-scale depthwise separable convolutional block to capture both the global receptive field and local positional features within the input series. We also devise dual forecasting heads, including one linear flatten head and one MLP flatten head. These forecasting heads jointly incorporate nonlinear and linear features to model future sequences independently. The prediction results from the dual heads are subsequently combined to produce the final prediction, denoted as $\hat{\boldsymbol{x}}$. Detailed explanations of these components will be provided in the following sections.

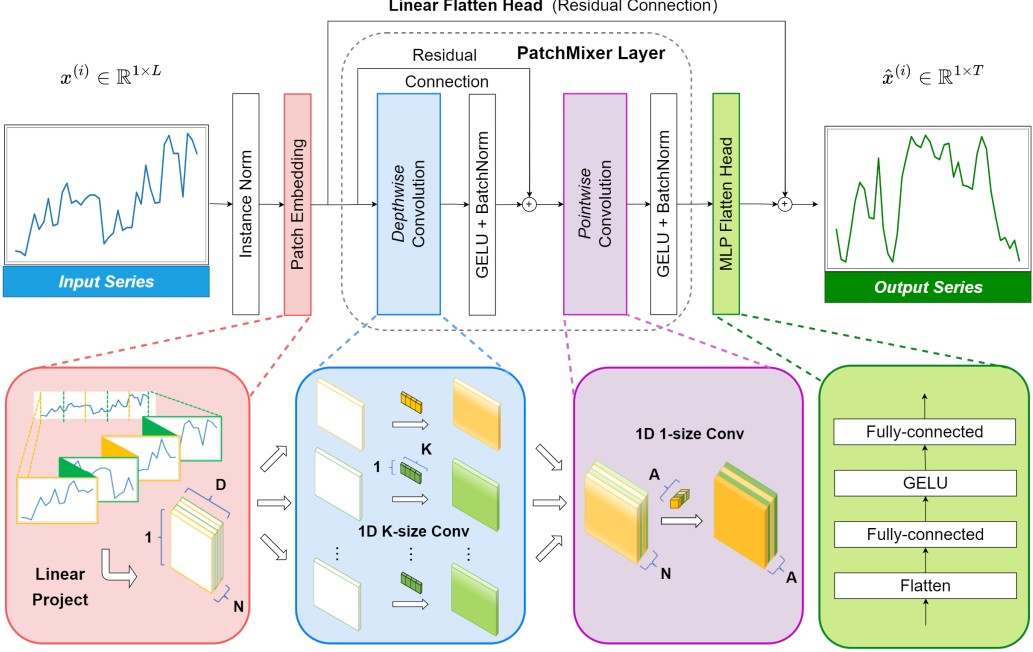

Figure 1: PatchMixer overview.

### 3.3 PATCH EMBEDDING

**Patch Representation.** Our work is inspired by PatchTST's (Nie et al., 2023) patch processing technique, which is proposed for use in the Transformer architecture. This method unfolds the input univariate time series $\mathbf{X}_{1D} \in \mathbb{R}^L$ through a sliding window with the length of $P$ and the step of $S$. Before transformation, it extends the original univariate time series $\mathbf{X}_{1D}$ by repeating its final value $S$ times. This process results in a series of 2D patches, maintaining their original relative positions. The patching process is illustrated by the following formulas.

$$\hat{\mathbf{X}}_{\text{2D}} = \text{Unfold}\left(\text{ReplicationPad}(\mathbf{X}_{\text{1D}}), \texttt{size=}P, \texttt{step=}S\right) \tag{1}$$

We hypothesize that the strong predictive performance observed in time series forecasting is attributed more to the patch embedding methods rather than the inherent predictive capabilities of the Transformers. Therefore, we design PatchMixer based on CNN architectures, which has shown to be faster at a similar scale with Transformers. For a fair comparison, we follow its setup of patch embedding and adopt default settings, specifically $P = 16$ and $S = 8$. This configuration results in a series of patches with a half-overlap between each patch. We further discussed the effects of overlaps and convolutional kernel sizes in the appendix section.

**Embedding without Positional Encoding.** Local positional information, signifying the temporal order of time series data, holds significant importance. However, the self-attention layer within the Transformer architecture is unable to inherently preserve this positional information. To augment the temporal context of time series inputs, traditional Transformer models such as Informer (Zhou et al., 2021), Autoformer (Wu et al., 2021), and FEDformer (Zhou et al., 2022) employ three types of input embeddings. This process is depicted in Equation 3, where *TFE* represents temporal feature encoding (for example, MinuteOfHour, HourOfDay, DayOfWeek, DayOfMonth, and MonthOfYear), *PE* represents position embedding, and *VE* represents value embedding.

$$\text{Embedding}(\mathbf{X}) = \text{sum}(TFE + PE + VE) : x^L \rightarrow x^D \tag{2}$$

Recent Transformers like PatchTST treat a patch as an input unit, eliminating the need for temporal feature encoding. Instead, they focus on capturing comprehensive semantic information that is not readily available at the point level. This is achieved by aggregating timesteps into subseries-level patches.

$$\text{Embedding}(\mathbf{X}) = \text{sum}(PE + VE) : x^{N \times S} \rightarrow x^{N \times D} \tag{3}$$

Unlike the Transformer, the CNN structure inherently possesses permutation variance, obviating the necessity of using position embedding in our model. Ultimately, our embedding can be represented by the following formula 4, which can be accomplished with a single linear layer.

$$\text{Embedding}(\mathbf{X}) = VE : x^{N \times S} \rightarrow x^{N \times D} \tag{4}$$

### 3.4 PATCHMIXER LAYER

As discussed in Section 2, previous CNNs in LTSF often modeled global relationships within time series data across multiple scales or numerous branches. In contrast, our Patchmixer employs single-scale depthwise separable convolution as the core module. The patch-mixing design separates the per-location (intra-patch) operations with depthwise convolution, and cross-location (inter-patch) operations with pointwise convolution, which allows our model to capture both the global receptive field and local positional features within the input series.

**Depthwise Convolution:** We use a specific type of grouped convolution where the number of groups equals the number of patches, denoted as $N$. To expand the receptive field, we employ a larger kernel size, typically equal to our default patch step, $S$, resulting in $K = 8$. In this process, each of the $N$ patches in the input feature map undergoes a separate convolution with one kernel. This operation generates $N$ feature maps, each corresponding to a specific patch. These feature maps are then concatenated sequentially to create an output feature map with $N$ channels. Depthwise convolution effectively employs group convolution kernels that are the same for patches sharing the same spatial locations. This allows the model to capture potential periodic patterns within the temporal patches. The following equation 5 shows the process of one univariate series $x^{N \times D}$ in layer $l - 1$ passing through the depthwise convolution kernel in layer $l$.

$$x_l^{N \times D} = \text{BatchNorm}\left(\sigma\{\texttt{Conv}_{N \rightarrow N}(x_{l-1}^{N \times D}, \texttt{stride=}K, \texttt{kernel\_size=}K)\}\right) \tag{5}$$

**Pointwise Convolution:** Our depthwise convolutional operation may not capture the inter-patch feature correlations effectively, which is why we follow it with pointwise convolution. Through this layer, we achieve temporal interaction between patches.

$$x_l^{N \times D} = \text{BatchNorm}\left(\sigma\{\texttt{ConvDepthwise}(x_{l-1}^{N \times D}\}) + x_{l-1}^{N \times D}\right) \tag{6}$$

$$x_{l+1}^{A \times D} = \text{BatchNorm}\left(\sigma\{\texttt{Conv}_{N \to A}(x_l^{N \times D}, \texttt{stride=1}, \texttt{kernel\_size=1})\}\right) \tag{7}$$

The above equations 6 and 7 demonstrate the process of the univariate series $x^{N \times D}$ in layer $l$ passing through the pointwise convolution kernel in layer $l + 1$, where $A$ means the number of output channels in pointwise convolution. Following each of these convolution operations, we apply an activation function and post-activation BatchNorm. In this context, $\sigma$ denotes an element-wise non-linearity activation function. For our work, we employ the GELU activation function as described in reference (Hendrycks & Gimpel, 2016).

We demonstrate the effectiveness of the separable convolution method, achieving superior overall performance compared to the attention mechanism. The details of the experiments are presented in Section 4.2. Additionally, pointwise convolution allows us to control the degree of information aggregation among patches by adjusting the number of output channels $A$, as illustrated in Figure 2. In the main text, we set $A = N$, which means the default setting is patch disaggregation. We delve into this character further in Appendix 5, indicating that patch aggregation can enhance prediction performance across various datasets.

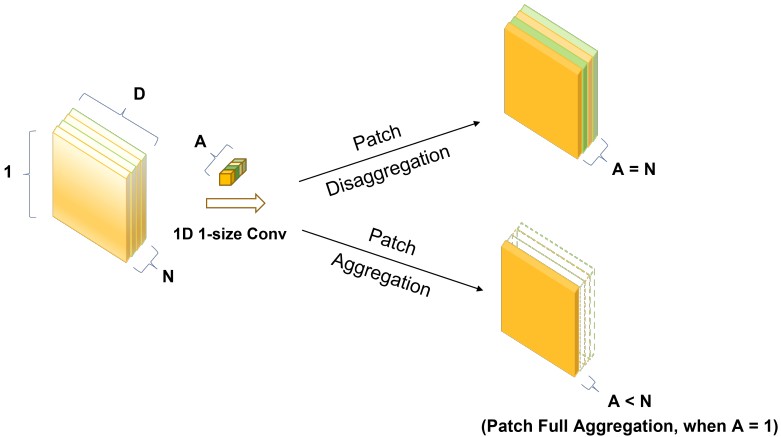

Figure 2: Patch Aggregation and Patch Disaggregation via Pointwise Convolution.

### 3.5 DUAL FORECASTING HEADS

Previous LTSF methods often followed a paradigm of decomposing inputs, such as employing the seasonal-trend decomposition technique and combining the decomposed components to obtain prediction results. Similarly, the multi-head attention mechanism in Transformers also involves decomposing and aggregating multiple outputs.

Motivated by the above instances, we propose a novel dual-head mechanism based on the decomposition-aggregation concept, one is dedicated to capturing linear features and the other focuses on capturing nonlinear variations. Specifically, PatchMixer extracts the overall trend of temporal changes through a linear residual connection spanning the convolution, and it uses an MLP forecasting head after a fully convolutional layer with a nonlinear function to meticulously fit the tiny changes in the prediction curve. Finally, we can derive the prediction results by summing their respective outputs. The utilization of dual heads yields a more effective mapping effect in comparison to the direct utilization of the previous single linear flattening head. We confirmed in Section 4.2 that both forecasting heads are indispensable for accurate prediction.

### 3.6 Normalization and Loss Fuction

**Instance Normalization.** This technique has recently been proposed to help mitigate the distribution shift effect between the training and testing data (Ulyanov et al., 2016; Kim et al., 2022). It simply normalizes each time series instance $\boldsymbol{x}^{(i)}$ with zero mean and unit standard deviation. In essence, we normalize each $\boldsymbol{x}^{(i)}$ before patching and the mean and deviation are added back to the output after dual forecasting heads.

**Loss Function.** Here we combine Mean Squared Error (MSE) and Mean Absolute Error (MAE) in an equal 1:1 ratio as our loss function. Surprisingly, we find that this simple method achieves superior accuracy overall, striking a balance between achieving lower MSE and MAE. The experimental details can be seen in Appendix A.2.

The MSE loss is:

$$\mathcal{L}_{\mathcal{MSE}} = \frac{1}{M} \sum_{i=1}^{M} \|\hat{\boldsymbol{x}}_{L+1:L+T}^{(i)} - \boldsymbol{x}_{L+1:L+T}^{(i)}\|_2^2, \tag{8}$$

while the MAE loss is:

$$\mathcal{L}_{\mathcal{MAE}} = \frac{1}{M} \sum_{i=1}^{M} \|\hat{\boldsymbol{x}}_{L+1:L+T}^{(i)} - \boldsymbol{x}_{L+1:L+T}^{(i)}\|. \tag{9}$$

## 4 Experiments

### 4.1 Multivariate Long-term Forecasting

**Datasets.** We evaluate the performance of our proposed PatchMixer on 7 commonly used long-term forecasting benchmark datasets: Weather, Traffic, Electricity, and 4 ETT datasets (ETTh1, ETTh2, ETTm1, ETTm2). The statistics of those datasets are summarized in Appendix A.1.1. It should be noted that ETTh1 and ETTh2 are small datasets, while ETTm1, ETTm2, and Weather are medium datasets. Traffic and Electricity each have more than 800 and 300 variables, respectively, with each variable containing tens of thousands of time points, naturally categorizing them as large datasets. Generally, smaller datasets contain more noise, while larger datasets exhibit more stable data distributions.

**Baselines and metrics.** We choose SOTA and representative LTSF models as our baselines, including Transformer-based models like PatchTST (2023), FEDformer (2022), Autoformer (2021), Informer (2021), in addition to two CNN-based models containing MICN (2023) and TimesNet (2023), with the significant MLP-based model DLinear (2023) to served as our baselines. To assess the performance of these models, we employ widely used evaluation metrics: MSE and MAE. The details of each baseline are described in Appendix A.1.2.

**Results.** Table 1 shows the multivariate long-term forecasting results. Our model outperforms all baseline methods significantly in all largest benchmarks, containing Traffic, Electricity, and Weather. On other datasets, we achieve the best performance across all or most prediction lengths. Quantitatively, PatchMixer demonstrates an overall relative reduction of **3.9**% on MSE and **3.0**% on MAE in comparison to the state-of-the-art Transformer (PatchTST). When evaluated against the best-performing MLP-based model (DLinear), our model showcases an overall decline of **11.6**% on MSE and **9.4**% on MAE. Moreover, in comparison to the achievable outcomes with the best CNN-based model (TimesNet), we demonstrate a remarkable overall relative reduction of **21.2**% on MSE and **12.5**% on MAE.

### 4.2 Ablation Study

**Training and Inference Efficiency.** We aim to demonstrate PatchMixer's superior efficiency in training and inference times compared to PatchTST, as shown in Figure 3. We conducted experiments using PatchTST's data loader and the ETTm1 dataset with a batch size of 8, resulting in data dimensions of $8 \times 7 \times L$ per batch. We report both inference time per batch and training time per epoch while varying the look-back length from 96 to 2880.

Table 1: Multivariate long-term forecasting results with our model PatchMixer. We use prediction lengths $T \in \{96, 192, 336, 720\}$ for all datasets. The best results are in **bold** and the second best results are in underlined.

| Models | Metrics | PatchMixer (Ours) MSE | MAE | PatchTST (2023) MSE | MAE | DLinear (2023) MSE | MAE | MICN (2023) MSE | MAE | TimesNet (2023) MSE | MAE | FEDformer (2022) MSE | MAE | Autoformer (2021) MSE | MAE | Informer (2021) MSE | MAE |
|---|---|---|---|---|---|---|---|---|---|---|---|---|---|---|---|---|---|
| Weather 96 | | **0.151** | **0.193** | 0.152 | 0.199 | 0.176 | 0.237 | 0.172 | 0.240 | 0.165 | 0.222 | 0.238 | 0.314 | 0.249 | 0.329 | 0.354 | 0.405 |
| Weather 192 | | **0.194** | **0.236** | 0.197 | 0.243 | 0.220 | 0.282 | 0.218 | 0.281 | 0.215 | 0.264 | 0.275 | 0.329 | 0.325 | 0.370 | 0.419 | 0.434 |
| Weather 336 | | **0.225** | **0.267** | 0.249 | 0.283 | 0.265 | 0.319 | 0.275 | 0.329 | 0.274 | 0.304 | 0.339 | 0.377 | 0.351 | 0.391 | 0.583 | 0.543 |
| Weather 720 | | **0.305** | **0.323** | 0.320 | 0.335 | 0.323 | 0.362 | 0.314 | 0.354 | 0.339 | 0.349 | 0.389 | 0.409 | 0.415 | 0.426 | 0.916 | 0.705 |
| Traffic 96 | | **0.363** | **0.245** | 0.367 | 0.251 | 0.410 | 0.282 | 0.479 | 0.295 | 0.593 | 0.321 | 0.576 | 0.359 | 0.597 | 0.371 | 0.733 | 0.410 |
| Traffic 192 | | **0.384** | **0.254** | 0.385 | 0.259 | 0.423 | 0.287 | 0.482 | 0.297 | 0.617 | 0.336 | 0.610 | 0.380 | 0.607 | 0.382 | 0.777 | 0.435 |
| Traffic 336 | | **0.393** | **0.258** | 0.398 | 0.265 | 0.436 | 0.296 | 0.492 | 0.297 | 0.629 | 0.336 | 0.608 | 0.375 | 0.623 | 0.387 | 0.776 | 0.434 |
| Traffic 720 | | **0.429** | **0.283** | 0.434 | 0.287 | 0.466 | 0.315 | 0.510 | 0.309 | 0.640 | 0.350 | 0.621 | 0.375 | 0.639 | 0.395 | 0.827 | 0.466 |
| Electricity 96 | | **0.129** | **0.221** | 0.130 | 0.222 | 0.140 | 0.237 | 0.153 | 0.264 | 0.168 | 0.272 | 0.186 | 0.302 | 0.196 | 0.313 | 0.304 | 0.393 |
| Electricity 192 | | **0.144** | **0.237** | 0.148 | 0.240 | 0.153 | 0.249 | 0.175 | 0.286 | 0.184 | 0.289 | 0.197 | 0.311 | 0.211 | 0.324 | 0.327 | 0.417 |
| Electricity 336 | | **0.164** | **0.257** | 0.167 | 0.261 | 0.169 | 0.267 | 0.192 | 0.303 | 0.198 | 0.300 | 0.213 | 0.328 | 0.214 | 0.327 | 0.333 | 0.422 |
| Electricity 720 | | **0.200** | **0.289** | 0.202 | 0.291 | 0.203 | 0.301 | 0.215 | 0.323 | 0.220 | 0.320 | 0.233 | 0.344 | 0.236 | 0.342 | 0.351 | 0.427 |
| ETTh1 96 | | **0.353** | **0.381** | 0.375 | 0.399 | 0.375 | 0.399 | 0.405 | 0.430 | 0.384 | 0.402 | 0.376 | 0.415 | 0.435 | 0.446 | 0.941 | 0.769 |
| ETTh1 192 | | **0.373** | **0.394** | 0.414 | 0.421 | 0.405 | 0.416 | 0.447 | 0.468 | 0.436 | 0.429 | 0.423 | 0.446 | 0.456 | 0.457 | 1.007 | 0.786 |
| ETTh1 336 | | **0.392** | **0.414** | 0.431 | 0.436 | 0.439 | 0.443 | 0.579 | 0.549 | 0.491 | 0.469 | 0.444 | 0.462 | 0.486 | 0.487 | 1.038 | 0.784 |
| ETTh1 720 | | **0.445** | **0.463** | 0.449 | 0.466 | 0.472 | 0.490 | 0.699 | 0.635 | 0.521 | 0.500 | 0.469 | 0.492 | 0.515 | 0.517 | 1.144 | 0.857 |
| ETTh2 96 | | **0.225** | **0.300** | 0.274 | 0.336 | 0.289 | 0.353 | 0.349 | 0.401 | 0.340 | 0.374 | 0.332 | 0.374 | 0.332 | 0.368 | 1.549 | 0.952 |
| ETTh2 192 | | **0.274** | **0.334** | 0.339 | 0.379 | 0.383 | 0.418 | 0.442 | 0.448 | 0.402 | 0.414 | 0.407 | 0.446 | 0.426 | 0.434 | 3.792 | 1.542 |
| ETTh2 336 | | **0.317** | **0.368** | 0.331 | 0.380 | 0.480 | 0.465 | 0.652 | 0.569 | 0.452 | 0.452 | 0.400 | 0.471 | 0.477 | 0.479 | 4.215 | 1.642 |
| ETTh2 720 | | 0.393 | 0.426 | **0.379** | **0.422** | 0.605 | 0.551 | 0.800 | 0.652 | 0.462 | 0.468 | 0.412 | 0.469 | 0.453 | 0.490 | 3.656 | 1.619 |
| ETTm1 96 | | 0.291 | 0.340 | **0.290** | 0.342 | 0.299 | 0.343 | 0.302 | 0.352 | 0.340 | 0.377 | 0.326 | 0.390 | 0.505 | 0.475 | 0.626 | 0.560 |
| ETTm1 192 | | **0.325** | **0.362** | 0.332 | 0.369 | 0.335 | 0.365 | 0.342 | 0.380 | 0.374 | 0.387 | 0.365 | 0.415 | 0.553 | 0.496 | 0.725 | 0.619 |
| ETTm1 336 | | **0.353** | **0.382** | 0.366 | 0.453 | 0.369 | 0.386 | 0.381 | 0.403 | 0.392 | 0.413 | 0.392 | 0.425 | 0.621 | 0.537 | 1.005 | 0.741 |
| ETTm1 720 | | **0.413** | **0.413** | 0.420 | 0.533 | 0.425 | 0.421 | 0.434 | 0.447 | 0.433 | 0.436 | 0.446 | 0.458 | 0.671 | 0.561 | 1.133 | 0.845 |
| ETTm2 96 | | 0.174 | 0.256 | **0.165** | **0.255** | 0.167 | 0.260 | 0.188 | 0.286 | 0.183 | 0.271 | 0.180 | 0.271 | 0.255 | 0.339 | 0.355 | 0.462 |
| ETTm2 192 | | 0.227 | 0.295 | **0.220** | **0.292** | 0.224 | 0.303 | 0.236 | 0.320 | 0.242 | 0.309 | 0.252 | 0.318 | 0.281 | 0.340 | 0.595 | 0.586 |
| ETTm2 336 | | **0.266** | **0.323** | 0.278 | 0.329 | 0.281 | 0.342 | 0.295 | 0.355 | 0.304 | 0.348 | 0.324 | 0.364 | 0.339 | 0.372 | 1.270 | 0.871 |
| ETTm2 720 | | **0.344** | **0.372** | 0.367 | 0.385 | 0.397 | 0.421 | 0.422 | 0.445 | 0.385 | 0.400 | 0.410 | 0.420 | 0.433 | 0.432 | 3.001 | 1.267 |

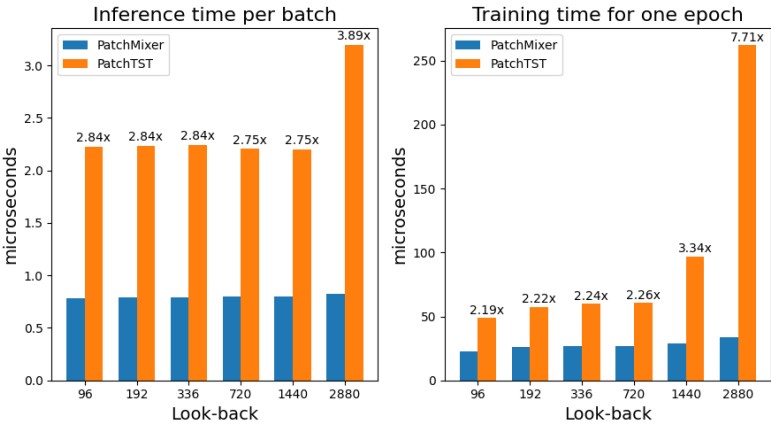

Figure 3: Comparison of Training and Inference Times: PatchMixer vs. PatchTST

Our results highlight two key improvements. First, PatchMixer achieves a 3x faster inference and 2x faster training speed compared to PatchTST. Second, PatchTST's performance is highly sensitive to the length of the look-back window, particularly when it reaches or exceeds 1440. In contrast, PatchMixer exhibits fewer fluctuations in both inference and training times with increasing historic length, contributing to higher accuracy and computational efficiency. All experiments in this subsection are conducted on the same machine, utilizing a single GPU RTX4090 for consistent and reliable findings. Moreover, we also explore where the speed-up comes from in Appendix A.5.

**Depthwise Separable Convolution vs. Self-attention Mechanism, Standard Convolution.** To assess the effectiveness of depthwise separable convolution, we replace the module in PatchMixer

Table 2: Ablation of depthwise separable convolution in the Traffic, ETTm1, and ETTm2 datasets. We replace the convolutional module with a transformer encoder in PatchTST and standard convolution. The better results are highlighted in **bold**.

| Datasets | | Traffic | | | | ETTm1 | | | | ETTh1 | | | |
|---|---|---|---|---|---|---|---|---|---|---|---|---|---|
| Prediction Length T | | 96 | 192 | 336 | 720 | 96 | 192 | 336 | 720 | 96 | 192 | 336 | 720 |
| Dual Heads | MSE | 0.377 | 0.391 | 0.399 | 0.429 | **0.290** | 0.327 | 0.356 | 0.420 | **0.355** | **0.373** | 0.392 | **0.440** |
| | MAE | 0.251 | 0.256 | 0.258 | 0.283 | **0.340** | 0.362 | 0.381 | 0.416 | 0.384 | 0.394 | 0.412 | **0.455** |
| Attention Mechanism | MSE | 0.368 | 0.388 | 0.401 | 0.447 | 0.294 | 0.331 | 0.360 | 0.422 | **0.355** | 0.378 | 0.393 | 0.451 |
| + Dual Heads | MAE | **0.240** | **0.249** | **0.256** | **0.273** | **0.340** | 0.365 | 0.386 | 0.417 | **0.382** | 0.397 | 0.411 | 0.460 |
| PatchMixer Layer | MSE | **0.362** | **0.382** | **0.392** | **0.428** | **0.290** | **0.325** | **0.353** | **0.413** | **0.355** | **0.373** | **0.391** | 0.446 |
| + Dual Heads | MAE | 0.242 | 0.252 | 0.257 | 0.282 | **0.340** | **0.361** | **0.382** | **0.413** | 0.383 | **0.394** | **0.410** | 0.463 |

with the transformer encoder of PatchTST and standard convolution separably. Each uses one layer and follows the same configuration.

The results are shown in Table 9, which implies that convolutional layers outperform attention layers in the majority of cases. The results of depthwise separable convolution are close to those of standard convolution, whereas standard convolution achieves its best results primarily in small and medium-sized datasets. In contrast, the superior predictive performance of separable convolution is evenly distributed across datasets of various sizes.

**Dual Forecasting Heads.** We use a single Linear Flatten Head as a baseline. In Figure 4, it is evident that the dual-head mechanism outperforms all other results and is at least comparable to one of the output heads within the dual-head setup. This outcome underscores the effectiveness of the dual-head mechanism when compared to a single-layer output head.

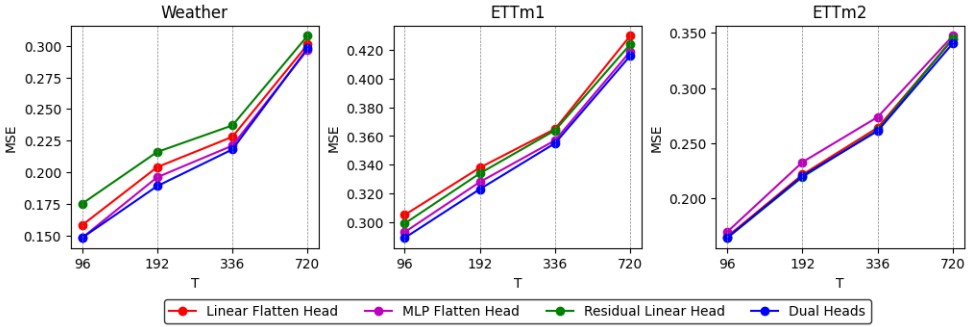

Figure 4: Ablation study of dual heads. We use prediction lengths $T \in \{96, 192, 336, 720\}$ in three datasets Weather, ETTm1, and ETTm2.

**Varying Look-back Window.** In principle, the large receptive field is beneficial for improving performance, while the receptive field of the look-back window in time series analysis is also important. Generally speaking, a powerful LTSF model with a strong temporal relation extraction capability should be able to achieve better results with longer input historical sequences. However, as argued in Zeng et al. (2023), this phenomenon has not been observed in most of the Transformer-based models. We also demonstrate in Figure 5 that in most cases, these Transformer-based baselines except PatchTST have not benefited from longer look-back window $L$, which indicates their ineffectiveness in capturing long-term temporal information. In contrast, recent baselines such as PatchTST, DLinear, and our PatchMixer consistently reduce the MSE scores as the receptive field increases, which confirms our model's capability to learn from the longer look-back window.

---

[1]We omit the results of Informer because its performance significantly deviated from the other models, which could distort the comparison results.

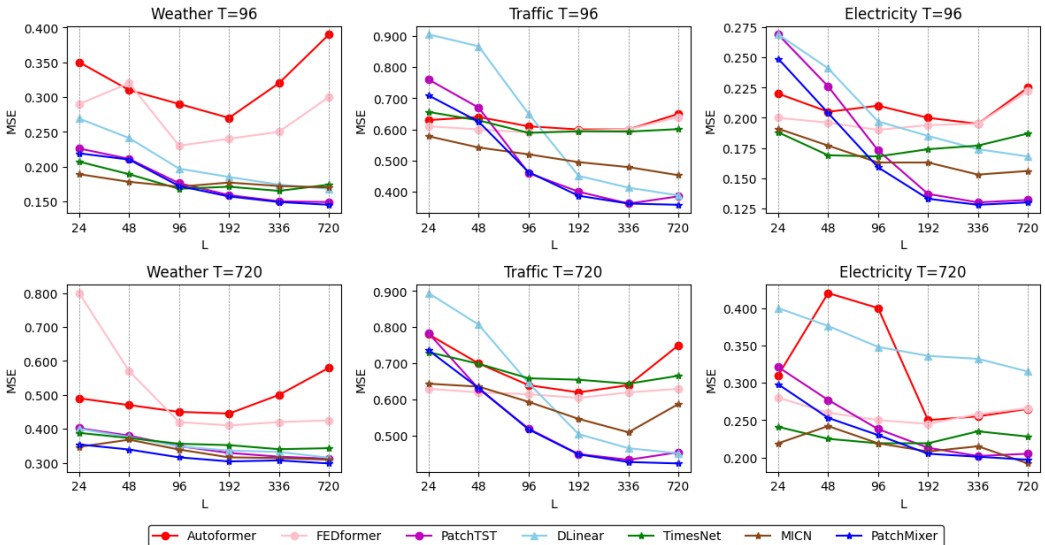

Figure 5: Forecasting performance (MSE) with varying look-back windows on 3 largest datasets: Traffic, Electricity, and Weather. The look-back windows are selected to be $L = 24, 48, 96, 192, 336, 720$, and the prediction horizons are $T = 96, 720$. We use our PatchMixer and the baselines for this experiment.[1]

## 5 CONCLUSION AND FUTURE WORK

In this paper, we introduce PatchMixer, a novel CNN-based model for long-term time series forecasting. PatchMixer leverages depthwise separable convolution with an innovative patch-mixing design to efficiently capture both global and local temporal patterns without self-attention mechanisms. We also highlight the importance of modeling linear and nonlinear components separately through dual forecasting heads, further enhancing our model's predictive capability. Our experiments demonstrate that PatchMixer outperforms state-of-the-art methods in terms of prediction accuracy while being significantly faster in both training and inference.

While our model has exhibited promising results, there is still potential for improvement, especially in the integration of external temporal features. Long-term time series forecasting often relies on external factors like holidays, weather conditions, or economic indicators. Effectively incorporating these features into patch-based models presents a challenge due to the inherently local nature of patch-based operations. These models tend to focus on individual time points rather than broader periods. We sincerely hope that further research in this direction could lead to more robust forecasting solutions.

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

## A  APPENDIX

### A.1  EXPERIMENTAL DETAILS

#### A.1.1  DATASETS

We use 7 popular multivariate datasets provided in (Wu et al., 2021) for forecasting. Detailed statistical data on the size of the datasets are as follows.

Table 3: Statistics of popular datasets used for benchmarking.

| Datasets | Weather | Traffic | Electricity | ETTh1 | ETTh2 | ETTm1 | ETTm2 |
|---|---|---|---|---|---|---|---|
| Variables | 21 | 862 | 321 | 7 | 7 | 7 | 7 |
| Timesteps | 52696 | 17544 | 26304 | 17420 | 17420 | 69680 | 69680 |
| Frequencies | 10 Minutes | 1 Hour | 1 Hour | 1 Hour | 1 Hour | 15 Minutes | 15 Minutes |

- Weather:[2] This dataset collects 21 meteorological indicators in Germany, such as humidity and air temperature.
- Traffic:[3] This dataset records the road occupancy rates from different sensors on San Francisco freeways.
- Electricity:[4] This is a dataset that describes 321 customers' hourly electricity consumption.
- ETT (Electricity Transformer Temperature):[5] These datasets are collected from two different electric transformers labeled with 1 and 2, and each of them contains 2 different resolutions (15 minutes and 1 hour) denoted with m and h. Thus, in total we have 4 ETT datasets: *ETTm1*, *ETTm2*, *ETTh1*, and *ETTh2*.

#### A.1.2  BASELINES

We choose SOTA and the most representative LTSF models as our baselines:

- PatchTST (Nie et al., 2023): the current SOTA LTSF model as of August 2023. It uses channel-independent and patch techniques and achieves the highest performance by utilizing the vanilla Transformer encoders.
- DLinear (Zeng et al., 2023): a highly insightful work that employs simple linear models and trend decomposition techniques, outperforming all Transformer-based models at the time. This work inspired us to reflect on the utility of Transformers in LTSF and indirectly led to the numerous births of MLP-based models in recent studies.
- MICN (Wang et al., 2023): another non-transformer model that enhances the performance of CNN models in LTSF through down-sampled convolution and isometric convolution, outperforming many Transformer-based models. This excellent work has been selected for oral presentation at ICLR 2023.
- TimesNet (Wu et al., 2023): it proposes a task-general backbone for time series analysis and achieves SOTA in five mainstream time series analysis tasks, including short- and long-term forecasting, imputation, classification, and anomaly detection before DLinear.
- FEDformer (Zhou et al., 2022): it employs trend decomposition and Fourier transformation techniques to improve the performance of Transformer-based models in LTSF. It was the best-performing Transformer-based model before DLinear.
- Autoformer (Wu et al., 2021): it combines trend decomposition techniques with an auto-correlation mechanism, inspiring subsequent work such as FEDformer.

---

[2]https://www.bgc-jena.mpg.de/wetter/
[3]https://pems.dot.ca.gov/
[4]https://archive.ics.uci.edu/ml/datasets/ElectricityLoadDiagrams20112014
[5]https://github.com/zhouhaoyi/ETDataset

- Informer (Zhou et al., 2021): it proposes improvements to the Transformer model by utilizing a sparse self-attention mechanism and generative-style decoder, inspiring a series of subsequent Transformer-based LTSF models. This work was awarded Best Paper at AAAI 2021.

Classical RNN-based and CNN-based models, such as DeepAR (Salinas et al., 2020) and LSTnet (Lai et al., 2018), have been demonstrated in previous works to be less effective than previous Transformer-based models in LTSF (Zhou et al., 2021; Wu et al., 2021). Therefore, we did not include them in our baselines. We also noted other excellent works recently, such as Crossformer (Zhang & Yan, 2023), TiDE (Das et al., 2023), and MTS-Mixers (Li et al., 2023). However, due to limited resources, we could only select the LTSF models that were most relevant to our work and most representative at each stage as our baselines.

The implementation of all baselines is from their respective code repository. We also adopt their default hyper-parameters to train the models to expect look-back windows. It is noted that default look-back windows for different baseline models could be different. For previous models, such as Informer, Autoformer, and FEDformer, the default look-back window is $L = 96$; and for recent PatchTST and DLinear, the default look-back window is $L = 336$. The reason for this difference is that previous Transformer-based baselines are easy to overfit when the look-back window is long while the latter tend to underfit recent models. However, this can lead to an underestimation of the baselines.

Meanwhile, PatchTST(Nie et al., 2023) reports two versions of models, PatchTST/64 for the look-back window $L = 512$ and PatchTST/42 for $L = 336$. Therefore, to compare the best performance of our model and all baselines, we report $L = 336$ for PatchMixer, PatchTST/42 for PatchTST, and the best result in $L = 24, 48, 96, 192, 336$ for the other baseline models by default. Thus it could be a strong baseline.

### A.1.3 Implementation Details

**Model Parameters.** For all benchmarks, our model contains only 1 PatchMixer layer and dimension of latent space $D = 256$ by default. The MLP head consists of 2 linear layers with GELU (Hendrycks & Gimpel, 2016) activation function: one projecting the hidden representation $D = 2 \times T$ for the forecasting length $T$, and another layer that projects it back to the final prediction target $D = T$. The linear head includes a flatten operation and a linear layer aims to project the embed vector directly from $N \times D$ to $T$. Dropout with probability $0.2$ is applied in the patch embedding for all experiments. Our method uses the ADAMw optimizer. The training process will be early stopped after ten epochs if there is no loss degradation on the valid set. All the experiments are repeated 5 times with different seeds, implemented in PyTorch, and conducted on NVIDIA RTX 4090 24GB GPUs. The code will be publicly available.

### A.2 More Results on Ablation Study

**Varying Patch Strides.** The patch overlap $O$ is related to the patch step size, and the relationship is $O = P - S$. We conduct this experiment in Figure 6 on two large datasets, Weather and Electricity. One observation is that the MSE score oscillated in a small range (between 0.003 and 0.001) with different choices of $S$, indicating the robustness of our model to patch overlaps.

**Varying Convolutional Kernel Sizes.** We have conducted the experiments in Figure 7 on two large datasets, Weather and Electricity. There was no significant pattern in the MSE score with different choices of K, the ideal patch overlap and convolution kernel size may depend on the datasets.

**Loss Function.** We study the effects of different loss functions in Table 4. We include PatchTST as the SOTA benchmark for the Transformer-based model. By comparing results with MSE, MAE, SmoothL1loss, MSE, and MAE accordingly. The motivation of patching is natural: Since LTSF tasks usually use these two metrics, MSE and MAE, previous time series prediction tasks typically used MSE as the loss function, only a few models (Liu et al., 2022a) use MAE as the loss function for training. Recent work has also employed SmoothL1loss (Lin et al., 2023) and we notice that SmoothL1loss is a type of loss function that attempts to combine the advantages of both MSE and MAE. This observation motivates us to explore a multi-task loss approach.

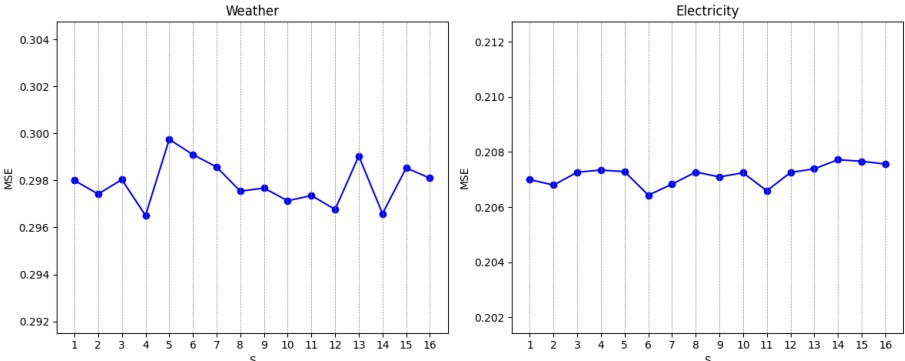

Figure 6: MSE scores with varying patch strides $S$ from 1 to 16 where the look-back window is 336 and the prediction length is 720.

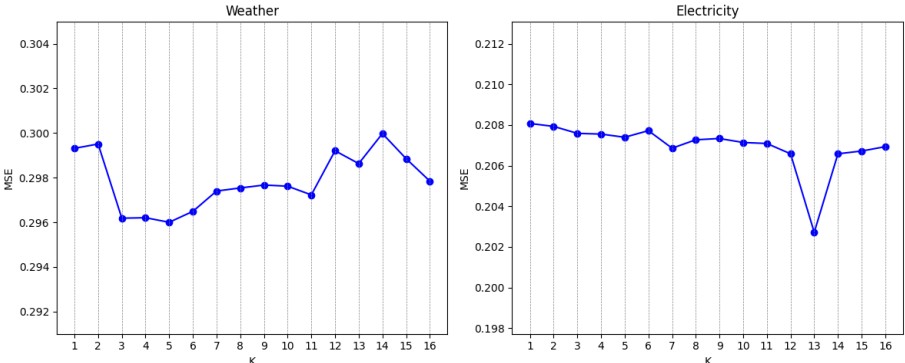

Figure 7: MSE scores with varying kernel sizes of depthwise convolution $K$ from 1 to 16 where the look-back window is 336 and the prediction length is 720.

In Table 4, we can intuitively observe that conventional training methods solely based on MSE do not yield optimal results, falling short of models trained solely on MAE or a combination of MSE and MAE (MSE+MAE). However, training exclusively on MAE tends to result in inferior MSE metrics. Taking all these factors into consideration, we ultimately decided to employ a training approach that combines MSE and MAE in a 1:1 ratio, aiming to strike a balance between these two loss functions to improve performance.

**Patch Aggregation vs Patch Disaggregation.** Intuitively, the relationships among patches are related to the potential period of the datasets, so fewer output channel numbers in pointwise convolution is beneficial for the model to learn periodic correlation through weight sharing, which is called patch aggregation. In the case of patch disaggregation, pointwise convolution can better fit the future development trend of time series by retaining more weights. Therefore, we can freely adjust the degree of patch aggregation by modifying the proportion of output channels and input channels in pointwise convolution.

From Table 5, we can see that for small and medium-sized datasets, patch aggregation has a significant improvement in prediction performance. It is noted that the results of full patch aggregation are similar to those using global average pooling. Both prevent overfitting by reducing the number of parameters, which is beneficial to improving the prediction accuracy of small and medium-sized data sets. However, for the two largest datasets, Traffic and Electricity, the effect of patch disaggregation is better. Moreover, we indicate that patch aggregation is a universal technique that can be used not only for PatchMixer but also for other models with patch presentation.

Table 4: Ablation study of loss functions for training in PatchMixer. 4 cases are included: (a) both MSE and MAE are included in loss function; (b) MSE; (c) MAE; (d) SmoothL1loss. The best results are in **bold**.

| Models | | PatchMixer | | | | | | | | PatchTST | | | | | | | |
|---|---|---|---|---|---|---|---|---|---|---|---|---|---|---|---|---|---|
| | | MSE+MAE | | MSE | | MAE | | SmoothL1loss | | MSE+MAE | | MSE | | MAE | | SmoothL1loss | |
| Metric | | MSE | MAE | MSE | MAE | MSE | MAE | MSE | MAE | MSE | MAE | MSE | MAE | MSE | MAE | MSE | MAE |
| Weather | 96 | **0.149** | 0.193 | 0.154 | 0.196 | 0.152 | 0.190 | 0.151 | 0.193 | 0.151 | 0.192 | 0.152 | 0.199 | 0.152 | 0.186 | 0.150 | 0.191 |
| | 192 | **0.191** | 0.233 | 0.197 | 0.237 | 0.193 | 0.231 | 0.194 | 0.237 | 0.195 | 0.234 | 0.197 | 0.243 | 0.197 | **0.230** | 0.196 | 0.236 |
| | 336 | **0.225** | 0.269 | **0.225** | 0.267 | 0.227 | **0.265** | 0.229 | 0.272 | 0.247 | 0.275 | 0.249 | 0.283 | 0.250 | 0.273 | 0.250 | 0.278 |
| | 720 | 0.307 | 0.324 | **0.302** | 0.322 | 0.308 | **0.321** | 0.309 | 0.326 | 0.321 | 0.328 | 0.320 | 0.335 | 0.322 | 0.326 | 0.321 | 0.330 |
| Traffic | 96 | **0.362** | 0.242 | 0.370 | 0.252 | 0.369 | 0.237 | 0.367 | 0.252 | 0.364 | 0.240 | 0.367 | 0.251 | 0.379 | **0.231** | 0.382 | 0.234 |
| | 192 | **0.382** | 0.252 | 0.388 | 0.258 | 0.388 | 0.244 | 0.386 | 0.256 | **0.382** | 0.245 | 0.385 | 0.259 | 0.398 | **0.239** | 0.403 | 0.245 |
| | 336 | **0.392** | 0.257 | 0.400 | 0.266 | 0.398 | 0.246 | 0.400 | 0.267 | 0.396 | 0.253 | 0.398 | 0.265 | 0.411 | 0.246 | 0.419 | 0.257 |
| | 720 | **0.428** | 0.282 | 0.436 | 0.288 | 0.429 | 0.266 | 0.435 | 0.290 | 0.434 | 0.277 | 0.434 | 0.287 | 0.443 | **0.265** | 0.460 | 0.296 |
| Electricity | 96 | **0.128** | 0.221 | **0.128** | 0.221 | **0.128** | **0.217** | 0.130 | 0.224 | 0.131 | 0.223 | 0.130 | 0.222 | 0.131 | 0.224 | 0.131 | 0.223 |
| | 192 | 0.144 | 0.237 | **0.142** | 0.236 | 0.143 | **0.233** | 0.145 | 0.240 | 0.148 | 0.239 | 0.148 | 0.240 | 0.149 | 0.241 | 0.148 | 0.240 |
| | 336 | 0.164 | 0.257 | 0.163 | 0.255 | **0.162** | **0.252** | 0.166 | 0.260 | 0.165 | 0.256 | 0.167 | 0.261 | 0.165 | 0.257 | 0.167 | 0.257 |
| | 720 | 0.201 | 0.290 | **0.199** | 0.289 | **0.199** | **0.284** | 0.204 | 0.293 | 0.208 | 0.293 | 0.202 | 0.291 | 0.207 | 0.290 | 0.207 | 0.290 |
| ETTh1 | 96 | 0.355 | 0.383 | 0.354 | 0.384 | **0.353** | 0.379 | 0.356 | 0.384 | 0.376 | 0.401 | 0.375 | 0.399 | 0.367 | 0.392 | 0.376 | 0.400 |
| | 192 | **0.373** | 0.394 | 0.376 | 0.397 | 0.376 | **0.392** | 0.375 | 0.394 | 0.411 | 0.418 | 0.414 | 0.421 | 0.411 | 0.416 | 0.412 | 0.418 |
| | 336 | **0.391** | **0.410** | 0.397 | 0.421 | 0.396 | **0.410** | 0.394 | 0.411 | 0.429 | 0.432 | 0.431 | 0.436 | 0.431 | 0.427 | 0.430 | 0.431 |
| | 720 | 0.446 | 0.463 | 0.446 | 0.462 | **0.437** | **0.450** | 0.444 | 0.462 | 0.445 | 0.462 | 0.449 | 0.466 | 0.443 | 0.455 | 0.442 | 0.460 |
| ETTh2 | 96 | **0.220** | 0.298 | 0.226 | 0.300 | 0.224 | **0.296** | 0.222 | 0.298 | 0.275 | 0.334 | 0.274 | 0.336 | 0.277 | 0.331 | 0.276 | 0.334 |
| | 192 | **0.267** | 0.332 | 0.276 | 0.335 | 0.272 | **0.331** | 0.270 | 0.333 | 0.340 | 0.375 | 0.339 | 0.379 | 0.343 | 0.374 | 0.341 | 0.375 |
| | 336 | **0.304** | **0.363** | 0.319 | 0.368 | 0.311 | 0.364 | 0.307 | 0.364 | 0.329 | 0.378 | 0.331 | 0.380 | 0.333 | 0.378 | 0.331 | 0.378 |
| | 720 | **0.375** | 0.417 | 0.395 | 0.427 | 0.380 | **0.416** | 0.377 | 0.417 | 0.378 | 0.419 | 0.379 | 0.422 | 0.382 | 0.417 | 0.380 | 0.419 |
| ETTm1 | 96 | 0.290 | 0.340 | 0.292 | 0.341 | 0.290 | **0.334** | **0.289** | 0.339 | 0.290 | 0.338 | 0.290 | 0.342 | 0.294 | 0.330 | 0.294 | 0.330 |
| | 192 | **0.325** | 0.361 | 0.326 | 0.362 | 0.328 | **0.357** | 0.327 | 0.362 | 0.334 | 0.365 | 0.332 | 0.369 | 0.339 | 0.359 | 0.337 | 0.358 |
| | 336 | **0.353** | 0.382 | 0.354 | 0.382 | 0.355 | **0.377** | 0.355 | 0.382 | 0.366 | 0.392 | 0.360 | 0.392 | 0.361 | 0.378 | 0.362 | 0.378 |
| | 720 | **0.413** | 0.413 | 0.417 | 0.413 | 0.415 | **0.409** | 0.416 | 0.413 | 0.421 | 0.420 | 0.420 | 0.424 | 0.415 | 0.414 | 0.415 | 0.414 |
| ETTm2 | 96 | 0.164 | 0.251 | 0.168 | 0.253 | 0.165 | 0.249 | **0.164** | 0.251 | 0.165 | 0.250 | 0.165 | 0.255 | 0.164 | **0.246** | 0.164 | 0.246 |
| | 192 | 0.220 | 0.291 | 0.224 | 0.291 | 0.219 | 0.285 | 0.219 | 0.289 | 0.219 | 0.289 | 0.220 | 0.292 | **0.215** | **0.283** | 0.218 | 0.285 |
| | 336 | 0.264 | 0.322 | 0.265 | 0.320 | 0.265 | 0.318 | **0.261** | 0.317 | 0.275 | 0.326 | 0.278 | 0.329 | 0.270 | 0.320 | 0.270 | 0.323 |
| | 720 | **0.342** | 0.375 | 0.343 | **0.370** | 0.347 | **0.370** | 0.345 | 0.371 | 0.365 | 0.382 | 0.367 | 0.385 | 0.355 | 0.374 | 0.363 | 0.380 |
| Avg. | | **0.292** | 0.316 | 0.296 | 0.318 | 0.294 | **0.312** | 0.294 | 0.318 | 0.305 | 0.322 | 0.306 | 0.327 | 0.307 | 0.318 | 0.309 | 0.322 |

## A.3 UNIVARIATE FORECASTING

Table 6 summarizes the results of univariate forecasting on ETT datasets. There is a target feature "oil temperature" within those datasets, which is the univariate series that we are trying to forecast. We cite the baseline results from (Zeng et al., 2023).

## A.4 ROBUSTNESS ANALYSIS

### A.4.1 RESULTS WITH DIFFERENT RANDOM SEEDS

The main tables in this article, including Table 1 and Table 6, are the averages of five random experiments. Besides, the remaining tables are generated using a fixed random number seed 2021. To examine the robustness of our results, we train the PatchMixer model with 5 different random seeds: 2021, 2022, 2023, 2024, and 2025. We calculate the MSE and MAE scores with each selected seed. The mean and standard derivation of the results are reported in Table 7. The variances are considerably small which indicates the robustness of our model.

### A.4.2 RESULTS WITH DIFFERENT MODEL PARAMETERS

To see whether PatchMixer is sensitive to the choice of different settings, we perform another experiment with varying model parameters. We vary the number of PatchMixer layers $L = \{1, 2, 3\}$ and select the model dimension $D = \{128, 256\}$. In total, there are 6 different sets of model hyper-parameters to examine. Figure 8 shows the datasets are robust to the choice of model hyper-parameters.

Table 5: Patch Aggregation Analysis. We use prediction lengths $T \in \{96, 192, 336, 720\}$. **PFA** means Patch Full Aggregation and **PDA** means Patch Dis-Aggregation, while the better results of them are in **bold**.

| Models | | PatchMixer | | | | | | PatchTST | |
|---|---|---|---|---|---|---|---|---|---|
| | | Pooling | | PFA | | PDA | | | |
| Metric | | MSE | MAE | MSE | MAE | MSE | MAE | MSE | MAE |
| Weather | 96 | 0.147 | 0.188 | **0.148** | **0.191** | 0.149 | 0.193 | 0.152 | 0.199 |
| | 192 | 0.187 | 0.229 | **0.189** | **0.230** | 0.191 | 0.233 | 0.197 | 0.243 |
| | 336 | 0.220 | 0.261 | **0.218** | **0.261** | 0.225 | 0.269 | 0.249 | 0.283 |
| | 720 | 0.295 | 0.315 | **0.298** | **0.318** | 0.307 | 0.324 | 0.320 | 0.335 |
| Traffic | 96 | 0.385 | 0.257 | 0.382 | 0.256 | **0.362** | **0.242** | 0.367 | 0.251 |
| | 192 | 0.402 | 0.265 | 0.397 | 0.262 | **0.382** | **0.252** | 0.385 | 0.259 |
| | 336 | 0.414 | 0.272 | 0.409 | 0.269 | **0.392** | **0.257** | 0.398 | 0.265 |
| | 720 | 0.443 | 0.291 | 0.436 | 0.284 | **0.428** | **0.282** | 0.432 | 0.287 |
| Electricity | 96 | 0.133 | 0.226 | 0.131 | 0.224 | **0.128** | **0.221** | 0.130 | 0.222 |
| | 192 | 0.149 | 0.244 | 0.144 | 0.237 | **0.144** | **0.237** | 0.148 | 0.240 |
| | 336 | 0.169 | 0.263 | 0.166 | 0.258 | **0.164** | **0.257** | 0.167 | 0.261 |
| | 720 | 0.209 | 0.295 | 0.202 | 0.289 | **0.201** | **0.290** | 0.202 | 0.291 |
| ETTh1 | 96 | 0.355 | 0.383 | 0.357 | 0.384 | **0.355** | **0.383** | 0.375 | 0.399 |
| | 192 | 0.376 | 0.396 | 0.380 | 0.399 | **0.373** | **0.394** | 0.414 | 0.421 |
| | 336 | 0.391 | 0.410 | 0.393 | 0.411 | **0.391** | **0.410** | 0.431 | 0.436 |
| | 720 | 0.445 | 0.457 | **0.442** | **0.456** | 0.446 | 0.463 | 0.449 | 0.466 |
| ETTh2 | 96 | 0.220 | 0.298 | **0.221** | **0.299** | 0.225 | 0.300 | 0.274 | 0.336 |
| | 192 | 0.267 | 0.332 | **0.269** | 0.335 | 0.275 | **0.334** | 0.339 | 0.379 |
| | 336 | 0.304 | 0.363 | **0.306** | **0.366** | 0.316 | 0.368 | 0.331 | 0.380 |
| | 720 | 0.375 | 0.417 | **0.379** | **0.420** | 0.397 | 0.427 | 0.379 | 0.422 |
| ETTm1 | 96 | 0.301 | 0.343 | **0.289** | **0.338** | 0.290 | 0.340 | 0.290 | 0.342 |
| | 192 | 0.336 | 0.363 | **0.323** | **0.358** | 0.325 | 0.361 | 0.332 | 0.369 |
| | 336 | 0.364 | 0.386 | 0.355 | **0.378** | **0.353** | 0.382 | 0.366 | 0.392 |
| | 720 | 0.428 | 0.416 | 0.416 | **0.408** | **0.413** | 0.413 | 0.420 | 0.424 |
| ETTm2 | 96 | 0.165 | 0.252 | **0.165** | **0.252** | 0.176 | 0.257 | 0.165 | 0.255 |
| | 192 | 0.220 | 0.289 | **0.220** | **0.291** | 0.227 | 0.295 | 0.220 | 0.292 |
| | 336 | 0.262 | 0.320 | **0.261** | **0.320** | 0.267 | 0.322 | 0.278 | 0.329 |
| | 720 | 0.341 | 0.373 | **0.341** | 0.373 | 0.344 | **0.372** | 0.367 | 0.385 |

Table 6: Univariate long-term forecasting results with PatchMixer. ETT datasets are used with prediction lengths $T \in \{96, 192, 336, 720\}$. The best results are in **bold** and the second best results are in underlined.

| Models | | PatchMixer (Ours) | | PatchTST (2023) | | DLinear (2023) | | MICN (2023) | | TimesNet (2023) | | FEDformer (2022) | | Autoformer (2021) | | Informer (2021) | |
|---|---|---|---|---|---|---|---|---|---|---|---|---|---|---|---|---|---|---|
| Metric | | MSE | MAE | MSE | MAE | MSE | MAE | MSE | MAE | MSE | MAE | MSE | MAE | MSE | MAE | MSE | MAE |
| ETTh1 | 96 | **0.054** | **0.179** | 0.055 | **0.179** | 0.056 | 0.180 | 0.062 | 0.198 | 0.056 | 0.182 | 0.079 | 0.215 | 0.071 | 0.206 | 0.193 | 0.377 |
| | 192 | **0.066** | **0.198** | 0.071 | 0.205 | 0.071 | 0.204 | 0.079 | 0.223 | 0.072 | 0.209 | 0.104 | 0.245 | 0.114 | 0.262 | 0.217 | 0.395 |
| | 336 | **0.078** | **0.220** | 0.081 | 0.225 | 0.098 | 0.244 | 0.093 | 0.243 | 0.086 | 0.229 | 0.119 | 0.270 | 0.107 | 0.258 | 0.202 | 0.381 |
| | 720 | 0.093 | 0.243 | 0.087 | 0.232 | 0.189 | 0.359 | 0.132 | 0.292 | **0.082** | **0.228** | 0.142 | 0.299 | 0.126 | 0.283 | 0.183 | 0.355 |
| ETTh2 | 96 | **0.119** | **0.268** | 0.129 | 0.282 | 0.131 | 0.279 | 0.131 | 0.282 | 0.136 | 0.286 | 0.128 | 0.271 | 0.153 | 0.306 | 0.213 | 0.373 |
| | 192 | **0.147** | **0.305** | 0.168 | 0.328 | 0.176 | 0.329 | 0.193 | 0.350 | 0.186 | 0.340 | 0.185 | 0.330 | 0.204 | 0.351 | 0.227 | 0.387 |
| | 336 | **0.166** | **0.332** | 0.185 | 0.351 | 0.293 | 0.437 | 0.194 | 0.355 | 0.220 | 0.373 | 0.231 | 0.378 | 0.246 | 0.389 | 0.242 | 0.401 |
| | 720 | **0.217** | **0.374** | 0.224 | 0.383 | 0.276 | 0.426 | 0.295 | 0.442 | 0.241 | 0.392 | 0.278 | 0.420 | 0.268 | 0.409 | 0.291 | 0.439 |
| ETTm1 | 96 | 0.027 | 0.123 | **0.026** | **0.121** | 0.028 | 0.123 | 0.030 | 0.131 | 0.029 | 0.127 | 0.033 | 0.140 | 0.056 | 0.183 | 0.109 | 0.277 |
| | 192 | 0.040 | 0.152 | **0.039** | **0.150** | 0.045 | 0.156 | 0.044 | 0.156 | 0.047 | 0.163 | 0.058 | 0.186 | 0.081 | 0.216 | 0.151 | 0.310 |
| | 336 | 0.055 | **0.177** | 0.053 | **0.173** | 0.061 | 0.182 | 0.063 | 0.186 | 0.080 | 0.214 | 0.084 | 0.231 | 0.076 | 0.218 | 0.427 | 0.591 |
| | 720 | 0.075 | 0.211 | **0.074** | **0.207** | 0.080 | 0.210 | 0.078 | 0.210 | 0.084 | 0.222 | 0.102 | 0.250 | 0.110 | 0.267 | 0.438 | 0.586 |
| ETTm2 | 96 | 0.067 | 0.188 | 0.065 | 0.186 | **0.063** | **0.183** | 0.064 | 0.184 | 0.066 | 0.187 | 0.067 | 0.198 | 0.065 | 0.189 | 0.088 | 0.225 |
| | 192 | 0.097 | 0.233 | 0.094 | 0.231 | **0.092** | **0.227** | 0.095 | 0.232 | 0.113 | 0.250 | 0.102 | 0.245 | 0.118 | 0.256 | 0.132 | 0.283 |
| | 336 | 0.122 | 0.267 | 0.120 | 0.265 | **0.119** | **0.261** | 0.122 | 0.265 | 0.133 | 0.277 | 0.130 | 0.279 | 0.154 | 0.305 | 0.180 | 0.336 |
| | 720 | 0.172 | 0.324 | **0.171** | 0.322 | 0.175 | **0.320** | 0.202 | 0.348 | 0.182 | 0.333 | 0.178 | 0.325 | 0.182 | 0.335 | 0.300 | 0.435 |

Table 7: Long-term forecasting results with different random seeds in PatchMixer.

| | L | PatchMixer (Multivariate) | | PatchMixer (Univariate) | |
|---|---|---|---|---|---|
| | Metric | MSE | MAE | MSE | MAE |
| Weather | 96 | 0.1509±0.0021 | 0.1934±0.0022 | - | - |
| | 192 | 0.1935±0.0027 | 0.2360±0.0028 | - | - |
| | 336 | 0.2246±0.0033 | 0.2675±0.0028 | - | - |
| | 720 | 0.3046±0.0026 | 0.3230±0.0019 | - | - |
| Traffic | 96 | 0.3634±0.0015 | 0.2447±0.0023 | - | - |
| | 192 | 0.3836±0.0012 | 0.2537±0.0017 | - | - |
| | 336 | 0.3931±0.0010 | 0.2583±0.0011 | - | - |
| | 720 | 0.4291±0.0051 | 0.2826±0.0051 | - | - |
| Electricity | 96 | 0.1285±0.0010 | 0.2208±0.0007 | - | - |
| | 192 | 0.1442±0.0008 | 0.2373±0.0007 | - | - |
| | 336 | 0.1643±0.0014 | 0.2569±0.0011 | - | - |
| | 720 | 0.1998±0.0015 | 0.2889±0.0010 | - | - |
| ETTh1 | 96 | 0.3530±0.0017 | 0.3812±0.0019 | 0.0543±0.0018 | 0.1794±0.0042 |
| | 192 | 0.3734±0.0020 | 0.3937±0.0023 | 0.0662±0.0004 | 0.1984±0.0008 |
| | 336 | 0.3921±0.0070 | 0.4136±0.0109 | 0.0779±0.0009 | 0.2196±0.0009 |
| | 720 | 0.4453±0.0020 | 0.4630±0.0016 | 0.0930±0.0031 | 0.2432±0.0034 |
| ETTh2 | 96 | 0.2254±0.0013 | 0.3004±0.0005 | 0.1188±0.0009 | 0.2684±0.0005 |
| | 192 | 0.2743±0.0010 | 0.3344±0.0009 | 0.1465±0.0025 | 0.3045±0.0012 |
| | 336 | 0.3168±0.0020 | 0.3676±0.0013 | 0.1662±0.0009 | 0.3319±0.0006 |
| | 720 | 0.3934±0.0037 | 0.4263±0.0012 | 0.2168±0.0025 | 0.3744±0.0023 |
| ETTm1 | 96 | 0.2911±0.0016 | 0.3395±0.0012 | 0.0266±0.0001 | 0.1228±0.0003 |
| | 192 | 0.3253±0.0013 | 0.3618±0.0007 | 0.0401±0.0004 | 0.1519±0.0003 |
| | 336 | 0.3529±0.0008 | 0.3822±0.0014 | 0.0549±0.0005 | 0.1769±0.0009 |
| | 720 | 0.4134±0.0035 | 0.4132±0.0006 | 0.0752±0.0012 | 0.2108±0.0039 |
| ETTm2 | 96 | 0.1739±0.0021 | 0.2558±0.0007 | 0.0665±0.0006 | 0.1875±0.0008 |
| | 192 | 0.2274±0.0041 | 0.2954±0.0024 | 0.0967±0.0015 | 0.2334±0.0012 |
| | 336 | 0.2661±0.0011 | 0.3229±0.0013 | 0.1220±0.0007 | 0.2666±0.0005 |
| | 720 | 0.3428±0.0016 | 0.3727±0.0004 | 0.1724±0.0016 | 0.3242±0.0025 |

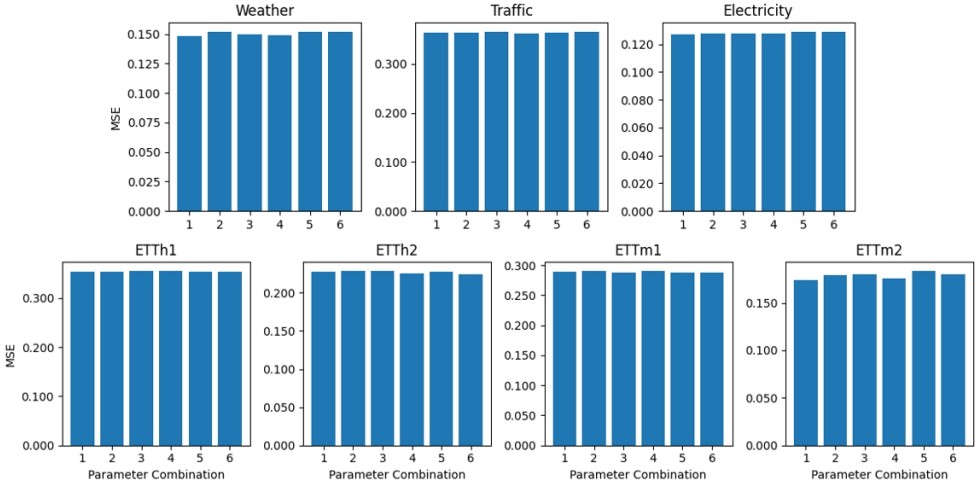

Figure 8: MSE scores with varying model parameters. Each bar indicates the MSE score of a parameter combination. The combinations $(L, D) = (1, 128), (1, 256), (2, 128), (2, 256), (3, 128), (3, 256)$ are orderly labeled from 1 to 6 in the figure. The model is run with PatchMixer to forecast 96 steps.

Table 8: Comparison of practical efficiency of methods under $L = 336$ and $T = 720$ on the ETTm1. MACs are the number of multiply-accumulate operations.

| Network Type | MACs | Computational Complexity |
|---|---|---|
| Attention Mechanism + Dual Heads | 293.63M | $O(N^2 \cdot D + N \cdot D^2)$ |
| Standard Convolution + Dual Heads | 175.57M | $O(N^2 \cdot D \cdot K)$ |
| Depthwise Separable Convolution + Dual Heads | 66.32M | $O(N \cdot D \cdot K + N \cdot D \cdot N)$ |

Table 9: Ablation of PatchMixer Layer in performances for multi-scale benchmarks, the large (Traffic), medium (ETTm1), and small-scale (ETTm2) datasets. The better results are highlighted in **bold** and the second best results are in underlined.

| Datasets | | Traffic | | | | ETTm1 | | | | ETTh1 | | | |
|---|---|---|---|---|---|---|---|---|---|---|---|---|---|
| Prediction Length T | | 96 | 192 | 336 | 720 | 96 | 192 | 336 | 720 | 96 | 192 | 336 | 720 |
| Attention Mechanism | MSE | 0.368 | 0.388 | 0.401 | 0.447 | 0.294 | 0.331 | 0.360 | 0.422 | 0.355 | 0.378 | 0.393 | 0.451 |
| + Dual Heads | MAE | **0.240** | **0.249** | **0.256** | **0.273** | 0.340 | 0.365 | 0.386 | 0.417 | 0.382 | 0.397 | 0.411 | **0.460** |
| Standard Convolution | MSE | 0.366 | 0.383 | 0.393 | **0.426** | **0.290** | **0.324** | 0.355 | **0.410** | **0.353** | **0.372** | 0.400 | **0.443** |
| + Dual Heads | MAE | 0.247 | 0.253 | 0.258 | 0.279 | **0.339** | **0.361** | **0.382** | 0.414 | **0.381** | 0.392 | 0.425 | 0.462 |
| Depthwise Separable Convolution | MSE | **0.362** | **0.382** | **0.392** | 0.428 | **0.290** | 0.325 | **0.353** | 0.413 | 0.355 | 0.373 | **0.391** | 0.446 |
| + Dual Heads | MAE | 0.242 | 0.252 | 0.257 | 0.282 | 0.340 | **0.361** | **0.382** | **0.413** | 0.383 | **0.394** | **0.410** | 0.463 |

## A.5 COMPUTATIONAL EFFICIENCY ANALYSIS

In this section, we explore the sources of acceleration and the efficiency of PatchMixer.

From a structural analysis, PatchMixer leverages the strength of convolutions for local modeling while overcoming the limitations of their receptive fields. Through patch embedding, which transforms the 1D sequence into a 2D matrix, we effectively address this limitation. A moderately sized kernel ($K = 8$) is sufficient to process stacked patches ($P = 16$), as the number of patches is treated as the channel dimension. This novel approach allows for feature extraction within and across patches, enabling simple 1D convolutions to capture temporal patterns both locally and globally.

Although both PatchMixer and PatchTST utilize parallelization, the computational cost under default configurations varies significantly. Assuming we set the number of patches to $N$, the embedding size to $D$, and the convolutional kernel size to $K$. PatchTST directly employs the vanilla Transformer's encoder, with the complexity primarily stemming from the self-attention mechanism and the feed-forward network, thus the total complexity is $O(N^2 \cdot D + N \cdot D^2)$. Conversely, PatchMixer's complexity mainly originates from depthwise separable convolutions. Its depthwise convolution's complexity is $O(N \cdot D \cdot K)$ and the pointwise convolution's complexity is $O(N \cdot D \cdot N)$. Therefore, the total complexity is the sum of these two. If a standard 1D convolution replaces the convolutional part of the PatchMixer Layer, the model's complexity becomes $O(N^2 \cdot D \cdot K)$.

We compare the average practical efficiencies with 5 runs. The PatchMixer Layer is composed of depthwise separable convolutions, so the PatchMixer model can be represented as Depthwise Separable Convolution + Dual Heads. Alternatively, a standard 1D convolution can replace the depthwise separable convolution, denoted as Standard Convolution + Dual Heads, while the attention mechanism can be represented as Attention Mechanism + Dual Heads. As shown in Table 8, using standard convolution results in a lower inference cost than employing the attention mechanism, and the efficiency and complexity of the depthwise separable convolution approach are superior to that of standard convolution.

To assess the effectiveness of depthwise separable convolution, we replace the module in PatchMixer with the transformer encoder of PatchTST and standard convolution separably. Each uses one layer and follows the same configuration. As shown in Table 9, convolution methods generally outperform the attention mechanism in terms of predictive performance. Moreover, depthwise separable convolution, with fewer parameters, exhibits similar predictive performance to standard convolution. This underscores the efficient predictive capacity of our patch-mixing design.

