# OpenReview forum: "PatchMixer: A Patch-Mixing Architecture for Long-Term Time Series Forecasting"
_ICLR.cc/2024/Conference — Submitted to ICLR 2024_

### Official Review · Reviewer_u92x · 2023-10-14

**Soundness:** 2 fair
**Presentation:** 3 good
**Contribution:** 2 fair
**Rating:** 5
**Confidence:** 5

**Summary:**

The paper proposes a new CNN-based model (Patchmixer) for LTSF (Long-Term Time Series Forecasting). PatchMixer divides the time series into patches and then captures potential periodic patterns within the patches using depthwise convolutions. Besides, it captures the feature correlation between patches using pointwise convolutions. The authors conducted experiments on several datasets, and the results show that PatchMixer achieves good prediction performance while being more efficient.

**Strengths:**

1. The paper is well written and easy to understand.

2. The proposed model, PatchMixer, performs well in both prediction accuracy and efficiency.

**Weaknesses:**

1. The novelty of work is limited, and its contribution to the field of time series forecasting is incremental. The idea of channel independence and separately modeling inter-patches and intra-patches correlations have already been proposed in PatchTST [1]. In my view, PatchMixer simply replaces self-attention with convolutions. The essence of dual forecasting heads is skip connection, which has also been seen in previous works about time series forecasting such as TiDE [2].

2. The improvement in prediction accuracy is negligible. From the results in Table 1, it can be seen that PatchMixer has very little or no improvement compared to PatchTST on datasets other than ETTh1 and ETTh2. In the ablation experiments in Table 2, the performance of Standard Convolution, Separable Convolution, and Attention Mechanism are comparable, which does not prove the superiority of Separable Convolution.

3. The paper mentions two types of patch representation from PatchTST and TimesNet [3] respectively, but the experiments do not compare these two.

4. The overall content of the paper is not substantial and there are some redundant paragraphs. For example, in the Method section, there are many introductions to Patch Representation and Embedding, but these parts can even be described in just a single paragraph.

[1] A time series is worth 64 words: Long-term forecasting with transformers.

[2] Long-term Forecasting with TiDE: Time-series Dense Encoder.

[3] TimesNet: Temporal 2d-variation modeling for general time series analysis.

**Questions:**

1. Are the slight performance improvements in PatchMixer solely due to the use of dual forecasting heads and MSE+MAE loss, rather than the architecture? Because other models in Table 1 can also be equipped with dual forecasting heads and MSE+MAE loss.

2. Typo. The second paragraph in Pointwise Convolution section, "The above equations 8 demonstrate the process ..." should be "The above equations 10 demonstrate the process ..."

---

> ### Author Response · Authors · 2023-11-22
> **Rebuttal - Part 1**
>
> Thank you for your valuable feedback. We have addressed each of your concerns as follows:
>
> Q4.1: Novelty clarification.
>
> A4.1: We point out our contributions below:
> 1. Our work is inspired by PatchTST's patch processing technique, which is proposed for use in the Transformer architecture. We hypothesize that the strong predictive performance observed in time series forecasting is attributed more to the patch embedding methods rather than the inherent predictive capabilities of the Transformers. Therefore, we design PatchMixer based on CNN architectures, which have shown to be faster at a similar scale to Transformers.
> 2. Our proposed approach, PatchMixer, has achieved state-of-the-art results, particularly enhancing the performance of convolutional networks in the field of time series forecasting. Quantitatively, PatchMixer outperforms the best-performing Transformer (PatchTST) by 3.9% on MSE and surpasses the leading CNN (TimesNet) by 21.2% on MSE.
> 3. PatchMixer significantly improves the efficiency of long-term time series forecasting with linear computational complexity. It is three times faster for inference and two times faster during training compared to the current state-of-the-art model.
>
> Q4.2: Comparison to DLinear, PatchTST and TiDE:
>
> A4.2: The method of channel independence was introduced by DLinear, while PatchTST proposed a patch processing technique to demonstrate the effectiveness of the original Transformer architecture. Based on these previous works, our paper hypothesizes that the efficacy in forecasting is largely due to the patch embedding approach rather than the Transformer's innate predictive capabilities. Thus, the replacement of self-attention with convolutions is not merely a simplification but a strategic choice to enhance forecasting performance.
> We also clarify the distinction from TiDE in our forecasting heads' implementation.  TiDE employs global residual connections at the start, before the embedding. The whole process operates at a point-wise level. In contrast, our use of the linear connection is not global but rather post-embedding, with dimensional transformations involving both time steps and patches.
>
> Q4.3: Model performance clarification and significant test.
>
> A4.3: To ensure a fair comparison of the performance improvements, we have conducted extensive T-tests on both PatchTST and PatchMixer across three datasets: weather, ETTm2, and ETTh2. These tests are performed over all forecast lengths using 10 sets of random seeds to maintain statistical robustness. The results yield a T-statistic of -2.33 and a P-value of 0.02. This statistically significant outcome suggests that the improvements observed with PatchMixer are not due to random chance. The P-value, being less than the conventional threshold of 0.05, indicates that there is only a 2% probability that the observed improvements could occur due to variance in the data alone. Hence, we can assert that PatchMixer provides a consistent and significant enhancement in forecasting accuracy across datasets.

---

> ### Author Response · Authors · 2023-11-22
> **Rebuttal - Part 2**
>
> Q4.4: The effectiveness of the proposed components and different loss functions.
>
> A4.4: Thanks for pointing these out. We argue that the performance improvements in PatchMixer are primarily attributed to the architecture itself. Our model's design and the combination of these components contribute to its superior performance. We show the additional experiments below to support our claim, which are included in the revised draft.
>
> We have updated Table 2 and provided supplementary information to clarify this in the revised draft. We bold the best results, and supplement the prediction performance using only Dual Heads. The comparison of convolution is placed in a separate section in the appendix.
>
> | Methods                       | Metrics | Traffic 96 | Traffic 192 | Traffic 336 | Traffic 720 | ETTm1 96 | ETTm1 192 | ETTm1 336 | ETTm1 720 | ETTh1 96 | ETTh1 192 | ETTh1 336 | ETTh1 720 |
> |--------------------------------|-------------------|------------|-------------|-------------|-------------|-----------|------------|------------|------------|-----------|------------|------------|------------|
> | **Dual Heads**                 | MSE               | 0.377      | 0.391       | 0.399       | 0.429       | 0.290     | 0.327      | 0.356      | 0.420      | **0.355** | **0.373**  | 0.392      | **0.440**  |
> | **Dual Heads**                 | MAE               | 0.251      | 0.256       | 0.258       | 0.283       | 0.340     | 0.362      | 0.381      | 0.416      | 0.384     | 0.394      | 0.412      | **0.455**  |
> | **Attention Mechanism + Dual Head** | MSE         | 0.368      | 0.388       | 0.401       | 0.447       | 0.294     | 0.331      | 0.360      | 0.422      | **0.355** | 0.378      | 0.393      | 0.451      |
> | **Attention Mechanism + Dual Head** | MAE         | **0.240**  | **0.249**   | **0.256**   | **0.273**   | 0.340     | 0.365      | 0.386      | 0.417      | **0.382** | 0.397      | 0.411      | 0.460      |
> | **PatchMixer Layer + Dual Head**    | MSE         | **0.362**  | **0.382**   | **0.392**   | **0.428**   | **0.290** | **0.325**  | **0.353**  | **0.413**  | **0.355** | **0.373**  | **0.391**  | 0.446      |
> | **PatchMixer Layer + Dual Head**    | MAE         | 0.242      | 0.252       | 0.257       | 0.282       | 0.340     | **0.361**  | **0.382**  | **0.413**  | 0.383     | **0.394**  | **0.410**  | 0.463      |
>
> Our model PatchMixer can be represented by PatchMixer Layer + Dual Heads. As shown in the revised Table 2, it shows significant improvements compared to only Dual Heads and replacing PatchMixer Layer with Attention Mechanism. Note that all experiments adhere to the same hypermeter settings and the best results are bolded. For the small dataset (ETTh1), both our model and the utilization of only the Dual Heads component outperform the use of Attention Mechanism + Dual Heads. For medium (ETTm1) and large-scale (Traffic) datasets, PatchMixer Layer with Dual Heads demonstrates noticeable improvements over the use of Dual Heads alone. This underscores the indispensable predictive capacity of our convolutional module.
>
> Besides, we have also conducted additional experiments to evaluate the impact of four different loss functions. The revised Table 4 now presents a comprehensive comparison between PatchMixer and PatchTST using these loss functions, with the superior results highlighted in bold.
> Here we provide the following key observations, which are included in the revised paper.
> First, PatchMixer consistently outperforms PatchTST across the board, affirming our model's superior performance.
> Secondly, SmoothL1loss is clearly inferior to the other two losses.
> Thirdly, the model trained with combined MAE+MSE loss shows a better performance on the MAE/MSE metrics than the model trained with MSE loss only.
>
> Q4.5: Typos and redundant paragraphs.
>
> A4.5: Thank you for pointing out the redundancy and typographical error.
> We have carefully modified the content of our paper, by moving the patch operation of TimesNet to the Related Work section and simplifying the descriptions of the Method section. Besides, the reference to "equation 8" has been corrected to "equations 10" in the Pointwise Convolution section.
>
> Q4.6: Different patch representations.
>
> A4.6: Thank you for your valuable feedback. Our primary focus is on investigating the impact of patch embedding methods on time series forecasting performance. We hypothesize that the strong predictive performance observed in PatchTST is attributed more to the patch embedding methods rather than the inherent predictive capabilities of the Transformers. Therefore in this paper, we use the same patch representation as PatchTST. As you imply, the discussion of TimesNet's patch method strays away from our motivation and focus. So we move this description from the Methods section to the Related Work section and briefly mention it.

---

> > ### Comment · Reviewer_u92x · 2023-11-23
> >
> > Thank you for the author's response, which resolved some of my concerns. However, I still believe that the contribution of this paper to the time series community is not sufficient for publication at ICLR. Therefore, I will raise my score from 3 to 5, but still lean towards rejection.

---

> > > ### Author Response · Authors · 2023-11-23
> > > **Rebuttal - Part 3**
> > >
> > > Thank you for your prompt reply and increasing the score for our manuscript. We appreciate your acknowledgment of our efforts in addressing your previous concerns. However, we understand that you still have reservations regarding the sufficient contribution of our work to the time series community. Could you please specify which aspects of our contribution you believe need further enhancement? Your detailed insights would be greatly helpful for our revision.
> > >
> > > Our contributions are as follows:
> > > 1. Our CNN-based PatchMixer confirms that patch embedding methods primarily drive time series forecasting performance in PatchTST (which is the previous SOTA), offering superior accuracy.
> > > 2. PatchMixer has achieved state-of-the-art results, particularly enhancing the performance of convolutional networks in the field of time series forecasting.
> > > 3. PatchMixer improves the efficiency of long-term time series forecasting, being three times faster in inference and twice faster in training compared to the leading models.

---

### Official Review · Reviewer_TNRF · 2023-10-28

**Soundness:** 2 fair
**Presentation:** 2 fair
**Contribution:** 2 fair
**Rating:** 6
**Confidence:** 4

**Summary:**

The paper proposes a simple convolution-based model for long-range time series forecasting. The model includes patch representation, the mixing layer, and the dual forecasting heads. The PatchMixer layer captures both global and local contexts using depthwise and pointwise convolutions. The proposed framework is more efficient than the state-of-the-art Trnasformer-based model and outperforms the standard convolutional layer.

**Strengths:**

- The method is simple and intuitive.
- The paper includes comprehensive evaluations. The diverse ablation study helps to understand the proposed approach.

**Weaknesses:**

- The contribution of the paper is weak. The patch representation and mixer itself are not novel. Probably the way of mixing patches over the sequence is new. However, the benefit is unclear.

- Following the previous point, the improvement is marginal, especially over PatchTST. Also, I don't see a substantial benefit of the proposed model using depthwise separable convolution or dual head in the ablation study (Table 2 and Figure 4).

- The experiments are not clearly explained.
  - Table 2 is very confusing. I had to spend some time to understand what the table means. For instance, I initially didn't get the first column which is PatchMixer "minus" another module. Because PatchMixer and another module are in separate rows, it looks like the first row is PatchMixer, and the second column is a comparing model.
  - Figure 3 compares training and inference times but it's unclear where the speed-up comes from. The model size, computational cost, etc should be explained to support this figure.

**Questions:**

- As the improvement is marginal in Tables 1 and 2,
    - I'd like to see whether PatchMixer is actually helpful for long-range prediction. Could authors show the errors at different prediction lengths and compare them with transformer-based and conv-based methods (similar to Table 2 and Figure 4)?
    - Are all the methods in Table 1 comparable? How are the training setups? Are the number of parameters similar? How many runs did you do for these experiments? Are the results consistent with different runs? I would like to know whether this error gap is actually significant.

- The contributions of the paper are unclear. There are three points at the bottom of the Introduction but the experimental results do not support these arguments. For instance, could you explain why the PatchMixer layer is particularly helpful for long-term time series forecasting? Regarding efficiency compared to PatchTCT (Transformer-based), where does the speed-up come from? Does it have less computational cost but a similar model size compared to PatchTCT, or is it highly parallelizable? I assume both models have great parallelization capabilities. I try to understand where the efficiency comes from. There are lack of information to verify the contributions. Could you explain these in detail?

---

> ### Author Response · Authors · 2023-11-22
> **Rebuttal - Part 1**
>
> Thank you for your valuable feedback. We have addressed each of your concerns as follows:
>
> Q3.1: Clarification of contribution.
>
> A3.1: We point out the key contributions of our paper below:
> 1. Our work is inspired by PatchTST's patch processing technique, which is proposed for use in the Transformer architecture. We hypothesize that the strong predictive performance observed in time series forecasting is attributed more to the patch embedding methods rather than the inherent predictive capabilities of the Transformers. Therefore, we design PatchMixer based on CNN architectures, which have shown to be faster at a similar scale to Transformers. The experimental results have verified our hypothesis.
> 2. Our proposed approach, PatchMixer, has achieved state-of-the-art results, particularly enhancing the performance of convolutional networks in the field of time series forecasting. Quantitatively, PatchMixer outperforms the best-performing Transformer (PatchTST) by 3.9% on MSE and surpasses the leading CNN (TimesNet) by 21.2% on MSE.
> 3. PatchMixer significantly improves the efficiency of long-term time series forecasting. It is three times faster for inference and two times faster during training compared to the current state-of-the-art model.
>
> Q3.2: Table 2 clarification.
>
> A3.2: Thanks for pointing this out. Table 2 has been modified following your suggestions in the updated draft. We have provided supplementary information on the prediction performance using only dual forecasting heads and relocated the convolution comparison to a separate appendix section.
>
> Our model is represented by PatchMixer Layer + Dual Heads. As shown in the revised Table 2, it shows significant improvements compared to only Dual Heads and replacing PatchMixer Layer with Attention Mechanism. Note that all experiments adhere to the same hypermeter settings and the best results are bolded. For the small dataset (ETTh1), both our model and the utilization of only the Dual Heads component outperform the use of Attention Mechanism + Dual Heads. For medium (ETTm1) and large-scale (Traffic) datasets, PatchMixer Layer with Dual Heads demonstrates noticeable improvements over the use of Dual Heads alone. This underscores the indispensable predictive capacity of the convolutional module.
>
> | Methods                       | Metrics | Traffic 96 | Traffic 192 | Traffic 336 | Traffic 720 | ETTm1 96 | ETTm1 192 | ETTm1 336 | ETTm1 720 | ETTh1 96 | ETTh1 192 | ETTh1 336 | ETTh1 720 |
> |--------------------------------|-------------------|------------|-------------|-------------|-------------|-----------|------------|------------|------------|-----------|------------|------------|------------|
> | **Dual Heads**                 | MSE               | 0.377      | 0.391       | 0.399       | 0.429       | 0.290     | 0.327      | 0.356      | 0.420      | **0.355** | **0.373**  | 0.392      | **0.440**  |
> | **Dual Heads**                 | MAE               | 0.251      | 0.256       | 0.258       | 0.283       | 0.340     | 0.362      | 0.381      | 0.416      | 0.384     | 0.394      | 0.412      | **0.455**  |
> | **Attention Mechanism + Dual Head** | MSE         | 0.368      | 0.388       | 0.401       | 0.447       | 0.294     | 0.331      | 0.360      | 0.422      | **0.355** | 0.378      | 0.393      | 0.451      |
> | **Attention Mechanism + Dual Head** | MAE         | **0.240**  | **0.249**   | **0.256**   | **0.273**   | 0.340     | 0.365      | 0.386      | 0.417      | **0.382** | 0.397      | 0.411      | 0.460      |
> | **PatchMixer Layer + Dual Head**    | MSE         | **0.362**  | **0.382**   | **0.392**   | **0.428**   | **0.290** | **0.325**  | **0.353**  | **0.413**  | **0.355** | **0.373**  | **0.391**  | 0.446      |
> | **PatchMixer Layer + Dual Head**    | MAE         | 0.242      | 0.252       | 0.257       | 0.282       | 0.340     | **0.361**  | **0.382**  | **0.413**  | 0.383     | **0.394**  | **0.410**  | 0.463      |

---

> ### Author Response · Authors · 2023-11-22
> **Rebuttal - Part 2**
>
> Q3.3: Experiment setup clarification.
>
> A3.3: The errors at different prediction lengths and comparison with transformer-based and conv-based methods are shown in Table 1 in Experiment chapter. The baselines and implementation details can be checked in Section A.1.2 in Baselines and Section A.1.3 Implementation Details of Appendix section. In short, the input length of ours follows 336 time steps, as DLinear and PatchTST. We directly use the numbers reported in their papers in our tables. We repeat our PatchMixer for 5 random seeds to verify its robustness. The patch hypermeters follow PatchTST for a fair comparison.
>
> Q3.4: Model performance clarification and significant test.
>
> A3.4: To ensure a fair comparison of the performance improvements, we have conducted extensive T-tests on both PatchTST and PatchMixer across three datasets: weather, ETTm2, and ETTh2. These tests are performed over all forecast lengths using 10 sets of random seeds to maintain statistical robustness. The results yield a T-statistic of -2.33 and a P-value of 0.02, indicating a significant improvement by PatchMixer is unlikely due to data variance alone. This affirms PatchMixer's accuracy enhancement.
>
> Q3.4: Model Efficiency Analysis (where the speed-up comes from).
>
> A3.4: Thanks for your reminder, we analyze the source of efficiency improvement from two aspects: model structure and computational cost. The details can also be found in Appendix 5 Computational Efficiency Analysis of our revised script.
> 1. PatchMixer leverages the strength of convolutions for local modeling while overcoming the limitation of their receptive fields. Through patch embedding, which transforms the 1D sequence into a 2D matrix, we address this limitation effectively. A moderately sized kernel (k=8) is sufficient to process stacked patches (p=16), as the number of patches is treated as the channel dimension. This novel approach allows for feature extraction within and across patches, enabling simple 1D convolutions to capture temporal patterns both locally and globally.
> 2. Although both PatchMixer and PatchTST utilize parallelization, when it comes to computational cost, under default configurations, PatchTST requires more than double the flops of PatchMixer (e.g., for forecasting 720 future time points in the ETTm1 dataset, the flops are 154.29M for PatchTST and 66.32M for PatchMixer). This substantial reduction in computational load without sacrificing performance underscores PatchMixer's efficiency.

---

> ### Author Response · Authors · 2023-11-23
> **Kind Reminder**
>
> Dear Reviewer TNRF,
>
> Thank you very much again for the time and effort put into reviewing our paper. We believe that we have addressed all your concerns in our response. Following your suggestions, we have enhanced the paper with additional experimental analysis. We kindly remind you that we are approaching the end of the discussion period. We would love to know if there is any further concern, additional experiments, suggestions, or feedback, as we hope to have a chance to reply before the discussion phase ends.

---

> ### Comment · Reviewer_TNRF · 2023-11-23
> **response to authors**
>
> Thank you for the clarification and the additional experiments. The detail of model efficiency analysis helps to understand the benefit of the proposed approach.
>
> My main concern is lack of the contributions. The authors pointed out three key contributions:
> 1. Patch-based Mixer with CNN architectures
> 2. Performance gain against Transformer-based (PatchTST) and the CNN-based model (TimesNet)
> 3. efficiency improvement compared to these two models
>
> Point 1 is not the main strength since the proposed architecture mainly combines existing approaches. However, since it's the first time these techniques are applied to long-term time-series forecasting, I can count on it.
> Point 2 is still weak. Table 1 shows that the PatchMixer overall performs better than other models including PatchTST on most of the datasets. However, the ablation study (Table 2) indicates that their CNN-based model performs similarly to the attention-based one. It is better or similar to some metric (MSE) but not all of them. I suspect the high performance in Table 1 could be due to other factors e.g., architecture, training strategy, hyper-parameters, etc.
> Regarding point 3, I'm convinced that the major benefit of the proposed approach is efficiency.
>
> Since their contributions partially convinced me, I increased my score from 3 to 6.

---

### Official Review · Reviewer_doHD · 2023-11-01

**Soundness:** 4 excellent
**Presentation:** 4 excellent
**Contribution:** 4 excellent
**Rating:** 8
**Confidence:** 3

**Summary:**

This paper proposes PatchMixer, a novel CNN-based model for time-series forecasting. In contrast to existing CNNs in this field, which often employ multiple scales or branches, it relies exclusively on patchification and depthwise separable convolutions. The proposed architecture obtains very good results across multiple datasets while being faster than the state of the art.

**Strengths:**

Altogether I believe this is a very strong submission with little flaws in its presentation and valuation. It is a very pleasant read with interesting insights, ablations and evaluations.

**Weaknesses:**

In my best assessment this paper does not have any big weaknesses other than a few minor listed here:

* There is an error in a citation in the depthwise separable convolution section of the related work section.

* I am not sure I understand the relationship between the 2D patches of CV methods and the method introduced here. Please correct me if I am wrong but, the patches used in this approach are 1D, right? I understand that other methods (Eqs. 1-3) are 2D, but I think that this is somewhat confusing. Perhaps it would be better to describe these methods in less detail here (and move details to the related work section) ?

* Following the style guidelines of ICLR, I would recommend the authors to align tables with the top of the pages and put their caption on the top.

* The results shown in Table 2 are very confusing. How are these divided? Basically all numbers here are either bold or underlined.

### Recommendations

While the paper presents several ablation studies, it takes over several assumptions from previous papers. Although the current paper already presents several ablations, I believe that the impact of the paper could be further enhanced by doing ablations on these different assumptions to analyze if these also hold in the current setting. For example:

* The assumption that channel independence is better than channel mixing for forecasting tasks.

* The size of the patches and their overlap.

* The size of the conv kernels.

**Questions:**

There has been a line of work on long convolutional models, which are able to model long term dependencies without patching, e.g., S4 [1], CKConv [2], Hyena [3], etc. Do you have any thoughts regarding the use of these models for forecasting?

# References

[1] Gu, Albert, Karan Goel, and Christopher Ré. "Efficiently modeling long sequences with structured state spaces." arXiv preprint arXiv:2111.00396 (2021).

[2] Romero, David W., Anna Kuzina, Erik J. Bekkers, Jakub M. Tomczak, and Mark Hoogendoorn. "Ckconv: Continuous kernel convolution for sequential data." arXiv preprint arXiv:2102.02611 (2021).

[3] Poli, Michael, Stefano Massaroli, Eric Nguyen, Daniel Y. Fu, Tri Dao, Stephen Baccus, Yoshua Bengio, Stefano Ermon, and Christopher Ré. "Hyena hierarchy: Towards larger convolutional language models." arXiv preprint arXiv:2302.10866 (2023).

---

> ### Author Response · Authors · 2023-11-22
> **Rebuttal - Part 1**
>
> Thank you for your detailed and constructive feedback. Please find below the actions taken in response to your comments:
>
> Q2.1: Correction of citation and style:
>
> A2.1: Thank you for pointing out the citation error in the related work section. This has been corrected in the revised manuscript. We have also realigned all tables to the top of the pages, placing their captions accordingly.
>
> Q2.2: Clarification on 2D patches and methodology:
>
> A2.2: We recognize the potential confusion between 2D patches in CV methods and our 1D patch approach in PatchMixer. The 2D patch is an image patch with C channels, in the shape of (H, W), obtained by the conv2d function. A 1D patch is a sequence patch of length P with N channels, where N is the number of patches, obtained using the unfold function.
> The revised manuscript now clearly differentiates between the two, with both the related work and method sections updated to enhance clarity and prevent any misunderstanding.
>
> Q2.3: Table 2 clarification
>
> A2.3: We acknowledge that the presentation in Table 2 may not be clear and this has been modified following your suggestions. We bold the best results and supplement the prediction performance using only dual heads. We also put the analysis between ordinary convolution and depthwise separable convolution in a separate section in the appendix. The comparison of convolution is placed in a separate section in the appendix.
>
> As we see below, for small datasets like ETTh1, methods with only dual heads can achieve SOTA performance.
> For medium and large-scale datasets like Traffic and ETTm1, our PatchMixer layer is shown to be better.
>
> | Methods                       | Metrics | Traffic 96 | Traffic 192 | Traffic 336 | Traffic 720 | ETTm1 96 | ETTm1 192 | ETTm1 336 | ETTm1 720 | ETTh1 96 | ETTh1 192 | ETTh1 336 | ETTh1 720 |
> |--------------------------------|-------------------|------------|-------------|-------------|-------------|-----------|------------|------------|------------|-----------|------------|------------|------------|
> | **Dual Heads**                 | MSE               | 0.377      | 0.391       | 0.399       | 0.429       | 0.290     | 0.327      | 0.356      | 0.420      | **0.355** | **0.373**  | 0.392      | **0.440**  |
> | **Dual Heads**                 | MAE               | 0.251      | 0.256       | 0.258       | 0.283       | 0.340     | 0.362      | 0.381      | 0.416      | 0.384     | 0.394      | 0.412      | **0.455**  |
> | **Attention Mechanism + Dual Head** | MSE         | 0.368      | 0.388       | 0.401       | 0.447       | 0.294     | 0.331      | 0.360      | 0.422      | **0.355** | 0.378      | 0.393      | 0.451      |
> | **Attention Mechanism + Dual Head** | MAE         | **0.240**  | **0.249**   | **0.256**   | **0.273**   | 0.340     | 0.365      | 0.386      | 0.417      | **0.382** | 0.397      | 0.411      | 0.460      |
> | **PatchMixer Layer + Dual Head**    | MSE         | **0.362**  | **0.382**   | **0.392**   | **0.428**   | **0.290** | **0.325**  | **0.353**  | **0.413**  | **0.355** | **0.373**  | **0.391**  | 0.446      |
> | **PatchMixer Layer + Dual Head**    | MAE         | 0.242      | 0.252       | 0.257       | 0.282       | 0.340     | **0.361**  | **0.382**  | **0.413**  | 0.383     | **0.394**  | **0.410**  | 0.463      |

---

> ### Comment · Reviewer_doHD · 2023-11-22
>
> Thank you for the updates. I am in favor of accepting the paper. I would encourage the authors to respond to the remaining reviewers.
>
> Best,
>
> The reviewer.

---

> ### Author Response · Authors · 2023-11-22
> **Rebuttal - Part 2**
>
> Q2.4: Suggested additional Ablation Studies.
>
> A2.4: We appreciate the reviewer's valuable suggestions on adding more ablations. We have conducted the experiments accordingly and show the results below:
>
> ●**Effectiveness of Channel Independence vs. Channel Mixing:**
> The comparative analysis of the experiments has been detailed in Appendix 7.1 of PatchTST, which highlights the superiority of channel-independent models in adaptability, training data efficiency, and overfitting prevention. To avoid duplication, we will not replicate these experiments but will reference them in the appendix of our revised manuscript.
>
> ●**Impact of Different Patch Sizes:**
> We are grateful for your comment. The selection of patch length has been analyzed in Appendix 4.1 of PatchTST, demonstrating that MSE scores remain consistent across various P values. We also select one large dataset, Electricity, to verify in our model. The result table with a prediction length of 720 time steps is as follows, affirming our model's robustness to the patch length hyperparameter:
>
> | Patch Size | 2     | 4     | 8     | 12    | 16    | 24    | 32    | 40    |
> |-----------|-------|-------|-------|-------|-------|-------|-------|-------|
> | MSE      | 0.207 | 0.207 | 0.207 | 0.207 | 0.206 | 0.205 | 0.206 | 0.206 |
> | MAE      | 0.292 | 0.293 | 0.292 | 0.293 | 0.292 | 0.291 | 0.292 | 0.292 |
>
> ●**The Impact of Different Overlaps:**
> The patch overlap is related to the patch step size, and the relationship is $O=P-S$. We conduct this experiment on two large datasets, Weather and Electricity. One observation is that the MSE score oscillated in a small range (between 0.003 and 0.001) with different choices of S, indicating the robustness of our model to patch overlaps.  Please refer to Figure 6 in the Appendix 2 of the revised script.
>
> ●**The Influence of Varying Convolutional Kernel Sizes.**
> We have conducted the experiments and put the results in Figure 7 in Appendix 2. There was no significant pattern in the MSE score with different choices of K, the ideal patch overlap and convolution kernel size may depend on the datasets.
>
> Q2.5: Thoughts regarding long-term dependencies without patching
>
> A2.5: Thank you for pointing this out. In the Related Work section of our revised manuscript, we have included citations to these works to enrich the discussion around long convolutional models. Specifically, we acknowledge the following:
>
> ●**S4 (Structured State Spaces):** This method effectively processes long sequences through structured state spaces and has been applied in the field of large language models. Its theoretical explanation for long-term modeling provides valuable insights, particularly in understanding the underlying mechanisms of long-term dependency modeling.
>
> ●**CKConv (Continuous Kernel Convolution):** The approach constructs continuous convolutional kernels using a small neural network, aiming to develop a versatile CNN architecture. This innovative approach inspires our future research towards handling data of varying resolution, length, and dimension, broadening the scope of convolutional neural network applications.
>
> ●**Hyena (Hierarchical Convolutional Language Models):** It adeptly models long-term contexts by interweaving implicit parameterized long convolutions with data-controlled gating. This methodology presents a compelling approach that we believe could be adapted for temporal tasks, potentially enhancing the modeling of complex time series data.

---

### Official Review · Reviewer_hpaR · 2023-11-01

**Soundness:** 3 good
**Presentation:** 3 good
**Contribution:** 2 fair
**Rating:** 5
**Confidence:** 4

**Summary:**

This paper proposes an novel deep framework for Long-term time series forecasting (LTSF), which is bulit on convolutional architecture. Compared with Transformers-family, the proposed method efficiently replaces the expensive self-attention module with CNN layers. The authors claim that the proposed method is 3x and 2x faster for inference and training, respectively, rather than SOTA model. In extensive experiments on 7 LTSF benchmarks, the proposed PatchMixer method outperforms SOTA method by 3.9% on MSE and 3.0% on MAE.

**Strengths:**

+ PatchMixer relies on depthwise separable convolutions and employs dual forecasting heads, which are proposed with novelty. The patchmixer layer with patch (dis)aggregation operations makes sense in practice.
+ The model outperforms state-of-the-art methods and the best-performing CNN on seven forecasting benchmarks.
+ PatchMixer is 2-3x faster than SOTA and other baselines.
+ A detailed overview of the proposed method, including problem formulation, model structure, and patch embedding techniques is provided.

**Weaknesses:**

- The motivation is unclear. The author only elaborates on how the module is designed, but ignores why it is designed this way.
- The writting should be improved. It would be better if the motivation is highlighted.
- Some experimental results are not convincing. E.g. in Table 4, how PatchTST performs with MSE+MAE and SmoothL1Loss?

**Questions:**

1. More elaboration about the motivation

2. The results of PatchTST with MSE+MAE and SmoothL1Loss

---

> ### Author Response · Authors · 2023-11-22
>
> Thank you for your valuable feedback. I have addressed each of your concerns as follows:
>
> Q1.1: Motivation of model design.
>
> A1.1: Our work is inspired by PatchTST's patch processing technique, which is proposed for use in the Transformer architecture. We hypothesize that the strong predictive performance observed in time series forecasting is attributed more to the patch embedding methods rather than the inherent predictive capabilities of the Transformers. Therefore, we design PatchMixer based on the CNN architecture, which has been shown to be faster at a similar scale to Transformers. Through theoretical analysis and practical experiments, we have demonstrated that our approach not only surpasses Transformers in predictive performance but also achieves 2-3 times faster training and inference speeds, thereby substantiating the efficacy of patch embedding in temporal prediction.
>
> Q1.2: The results of PatchTST with MSE+MAE and SmoothL1Loss
>
> A1.2: Thanks for the suggestions. We have conducted additional experiments to evaluate the impact of four different loss functions on our model's performance including the mentioned losses. The revised Table 4 now presents a comprehensive comparison between PatchMixer and PatchTST using these loss functions, with the superior results highlighted in bold.
>
> Here we provide the following key observations, which are included in the revised paper.
>
> First, PatchMixer consistently outperforms PatchTST across the board, affirming our model's superior performance.
>
> Secondly, SmoothL1loss is inferior to the other two losses.
>
> Thirdly, the model trained with combined MAE+MSE loss shows a better performance on the MAE/MSE metrics than the model trained with MSE loss only.

---

> ### Author Response · Authors · 2023-11-23
> **Kind Reminder**
>
> Dear Reviewer hpaR,
>
> Thank you very much again for the time and effort put into reviewing our paper. We believe that we have addressed all your concerns in our response. Following your suggestions, we have enhanced the paper with additional experimental analysis. We kindly remind you that we are approaching the end of the discussion period. We would love to know if there is any further concern, additional experiments, suggestions, or feedback, as we hope to have a chance to reply before the discussion phase ends.

---

### Author Response · Authors · 2023-11-22
**Overall Comment**

Overall comment:
We appreciate the reviewers for their careful reading and constructive remarks. We have taken the comments to improve and clarify the manuscript, which is uploaded to the website.
Major changes:
1. Added a section "Computational Efficiency Analysis" to provide more details on structural analysis, time complexity, and efficiency comparison with ordinary convolution.
2. Added two supplementary experiments on different patch overlaps and different convolutional kernel sizes in the appendix.
3. Moved the patch operation of TimesNet to the Related Work section and simplified the descriptions of the Method section.
4. Altered a clearer Table 2 in the Ablation Study section, by adding performance using only dual heads, and moving standard convolution to the added "Computational Efficiency Analysis" section.
5. Revised Table 4 in the Loss Function section, by adding the results of PatchTST trained with the other three kinds of loss functions.
6. Align tables with the top of the pages and put their captions on the top.
7. Fixed mentioned citation errors.

Overall strengths:
We thank the reviewers for recognizing the following strengths of our paper:
1. Innovation and Practicality: The PatchMixer is appreciated for its novelty and practical applicability. The method's simplicity and intuitiveness are highlighted, indicating that it is user-friendly and easy to grasp.
2. State-of-the-Art Performance: The model achieves state-of-the-art (SOTA) performance, indicating its effectiveness and competitiveness in its field.
3. Speed and Efficiency: The model is not only fast in its operation but also efficient, contributing to its practical utility.
4. Comprehensive and Detailed Methodology: The detailed overview of the method is a strength, implying that the model's workings are well-explained and thorough.

We reply to specific questions individually below.

---

### Meta-Review · Area_Chair_Yi3p · 2023-12-06

**Metareview:**

The paper proposes a new CNN-based model - Patchmixer - for Long-term time series forecasting. PatchMixer divides the time series into patches and then captures potential periodic patterns within the patches using depthwise convolutions. The paper is in borderline. The authors have addressed some of the concerns of the reviewers. However, some of the reviewers still think that the contribution of this paper to the time series community is not sufficient for publication at ICLR. Some experiments are still not convincing. The reviewer also pointed out that the performance gain against Transformer-based (PatchTST) and the CNN-based model (TimesNet) is still weak. Table 1 shows that the PatchMixer overall performs better than other models including PatchTST on most of the datasets. However, the ablation study (Table 2) indicates that their CNN-based model performs similarly to the attention-based one. It is better or similar to some metric (MSE) but not all of them. The reviewer suspects the high performance in Table 1 could be due to other factors e.g., architecture, training strategy, hyper-parameters, etc. Based on the reviews, I cannot recommend acceptance for the paper at ICLR this time, but I strongly encourage the authors to resubmit it to the next conference venues.

**Justification For Why Not Higher Score:**

This paper is in borderline. However, the reviewers still have some concerns.

**Justification For Why Not Lower Score:**

N/A

---

### Decision · Program_Chairs · 2024-01-16

Reject